# Polynomial Composition Activations: Unleashing the Dynamics of Large Language Models

**Zhijian Zhuo**[1,2]* **Ya Wang**[2]* **Yutao Zeng**[2]† **Xiaoqing Li**[3] **Xun Zhou**[2] **Jinwen Ma**[1]†
[1] School of Mathematical Sciences, Peking University
[2] Seed-Foundation-Model, ByteDance
[3] Capital University of Economics and Business

## Abstract

Transformers have found extensive applications across various domains due to their powerful fitting capabilities. This success can be partially attributed to their inherent nonlinearity. Thus, in addition to the ReLU function employed in the original transformer architecture, researchers have explored alternative modules such as GELU and SwishGLU to enhance nonlinearity and thereby augment representational capacity. In this paper, we propose a novel category of polynomial composition activations (PolyCom), designed to optimize the dynamics of transformers. Theoretically, we provide a comprehensive mathematical analysis of PolyCom, highlighting its enhanced expressivity and efficacy relative to other activation functions. Notably, we demonstrate that networks incorporating PolyCom achieve the ***optimal approximation rate***, indicating that PolyCom networks require minimal parameters to approximate general smooth functions in Sobolev spaces. We conduct empirical experiments on the pre-training configurations of large language models (LLMs), including both dense and sparse architectures. By substituting conventional activation functions with PolyCom, we enable LLMs to capture higher-order interactions within the data, thus improving performance metrics in terms of accuracy and convergence rates. Extensive experimental results demonstrate the effectiveness of our method, showing substantial improvements over other activation functions. Code is available at https://github.com/BryceZhuo/PolyCom.

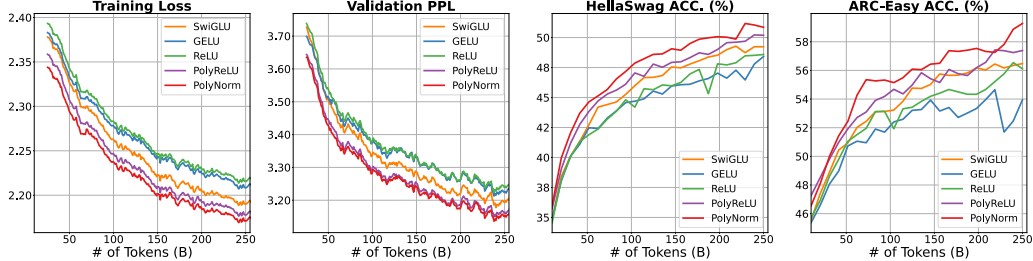

Figure 1: Training loss, validation perplexity (PPL), and downstream performance of 1B dense models. We compare models employing different activation functions, including SwiGLU, GELU, ReLU, PolyReLU, and PolyNorm. It indicates that models using PolyReLU and PolyNorm exhibit lower training loss and validation PPL, alongside better downstream performance.

## 1 Introduction

Transformers (Vaswani et al., 2017) have revolutionized the field of deep learning, facilitating unprecedented advancements in natural language processing (Radford et al., 2019; Zeng et al., 2020; Li et al., 2024; Zuo et al., 2024), computer vision (Dosovitskiy et al., 2021; Wang et al., 2022), and

---

*Equal contribution.
†Corresponding authors: Yutao Zeng (yutao.zeng@outlook.com) and Jinwen Ma (jwma@math.pku.edu.cn).

beyond (Dong et al., 2018; Arnab et al., 2021; Wang et al., 2020). Characterized by their attention mechanisms, transformers excel at capturing intricate relationships within data, making them indispensable in contemporary machine learning applications. However, despite their widespread success, there remain opportunities for further refinement, particularly concerning the selection of activation functions. The activation function plays a crucial role in determining the output of each neuron within a neural network. Traditionally, simple nonlinearities such as Rectified Linear Unit (ReLU) (Nair & Hinton, 2010) and its variants (Hendrycks & Gimpel, 2016; Krotov & Hopfield, 2016; Li et al., 2019; So et al., 2021) have been favored due to their computational efficiency and ease of implementation. Although effective, these activation functions are inherently limited in their ability to model complex higher-order relationships within data. This limitation can be particularly restrictive in transformer architectures, where the ability to capture subtle and complex dependencies is essential.

In this paper, we introduce a novel category of polynomial composition activation functions (**PolyCom**), specifically engineered to enhance the performance of transformer architectures. In contrast to conventional activation functions, which are predominantly linear or piecewise linear, polynomial composition activations facilitate the modeling of more complex patterns within data. This augmentation in the activation function's expressiveness endows the model with superior expressive capacity, enabling it to capture higher-order interactions that might otherwise be neglected. Unlike other forms of polynomials ((Hornik et al., 1989; Trefethen, 2019)) that suffer from inadequate approximation, exploding values, and oscillatory behavior, we demonstrate that PolyCom possesses a more potent expressive capability than both ReLU and traditional polynomials and achieves optimal approximation within Sobolev space.

We posit that the integration of polynomial composition activations within transformer models can lead to enhanced performance in tasks requiring intricate data interpretation. To evaluate this hypothesis, we conducted comprehensive experiments on the pre-training configurations of large language models (LLMs), including both dense and sparse architectures. These evaluations were performed across various benchmarks, assessing the performance of transformers employing polynomial composition activations in comparison to those utilizing traditional activation functions. The results indicate that the proposed method not only improves the model accuracy but also accelerates convergence rates, thereby suggesting that polynomial composition activations provide a substantive advantage in deep learning applications.

The main contributions of this paper are summarized in the following.

- We propose a new activation function PolyCom which is a composition of the polynomial and other types of function. In particular, we introduce two instances of PolyCom: PolyReLU and PolyNorm, and detail its integration into the transformer architecture.

- Theoretically, we derive bounds on the number of trainable parameters required for PolyReLU networks to approximate ReLU networks, and vice versa. Additionally, we show that a PolyReLU network of size $O(\epsilon^{-d/n})$ can approximate any function in Sobolev spaces with error tolerance $\epsilon$, achieving ***optimal approximation rates***.

- Empirically, we validate the effectiveness of this new activation function on LLMs with both 1B dense models and MoE models with 1B active and 7B total parameters. The results of both models demonstrate that PolyCom can accelerate the converging speed and significantly outperform SwiGLU, GELU, and ReLU *et al.*

The outline of this paper is structured as follows: In Section 2, we present the mathematical formulation of PolyCom and discuss its integration within transformer architectures. Section 3 delivers a comprehensive theoretical analysis of PolyCom, emphasizing its enhanced expressivity and effectiveness. In Section 4, we provide a detailed account of our experimental results involving large language models (LLMs). Section 5 provides an overview of related work in the field of activation functions and their applications in transformer models. Finally, we conclude the paper and outline potential directions for future research.

## 2 POLYNOMIAL COMPOSITION ACTIVATION FUNCTION

In this section, we present the mathematical formulation of the polynomial composition activation function (PolyCom) and detail its integration into the transformer architecture.

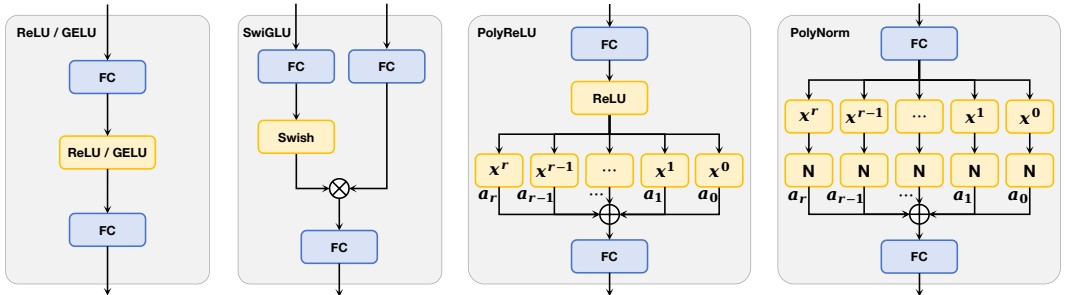

Figure 2: Block diagrams of Transformer MLP blocks utilizing ReLU/GELU, SwiGLU, PolyReLU and PolyNorm. "FC" stands for Fully Connected layer. "$x^i$" represents the $i$-th power of the input tensor $x$, "$a_j$" denotes the $j$-th element of the learnable weight vector $a$, "N" indicates a normalization operation.

**PolyCom.** The study of the polynomial activation function can be traced back to the seminal work of Hornik et al. (1989), which showed that neural networks with polynomial activation are not dense within the space of continuous functions. Additionally, empirical evidence has shown that deep neural networks employing pure polynomial activations tend to underperform (Trefethen, 2019). To overcome these limitations, we propose PolyCom, a novel composition of polynomials and other functions. Specifically, we explore two composition approaches

$$\begin{cases} \text{Type I:} & x \mapsto \sum_{i=0}^{r} a_i \rho^i(x), \\ \text{Type II:} & x \mapsto \sum_{i=0}^{r} a_i \rho(x^i), \end{cases} \quad a_i \in \mathbb{R}, \tag{1}$$

where $r \in \mathbb{N}$ denotes the order of PolyCom and $\rho$ represents an arbitrary function such as ReLU, PReLU, Sigmoid, SiLU, or normalization. The key distinction between the two approaches lies in whether the function is composed before or after the power operation. The distinction between the two approaches lies in whether the composition or the power is performed first. In the practical implementation of PolyCom, we use 3-order PolyCom ($r = 3$) with trainable coefficients $a_i$. For initialization, we set $a_i = 1/r$ for $i = 1, 2, \ldots, r$ and $a_0 = 0$. Our experiments on large language models (LLMs) show that 3-order PolyCom can indeed achieve extraordinary performance.

For Type I PolyCom, we specifically consider a composition involving the ReLU function due to its simplicity, which we term PolyReLU. An $r$-order PolyReLU is defined as

$$\text{PolyReLU}(x) = \sum_{i=0}^{r} a_i \text{ReLU}^i(x), \tag{2}$$

where $\text{ReLU}^i(x) = \max\{x, 0\}^i$. This formulation can be seen as an extension of both ReLU and square ReLU.

For Type II PolyCom, we introduce PolyNorm, which normalizes the powers to ensure consistent magnitudes across terms

$$\text{PolyNorm}(\boldsymbol{x}) = \sum_{i=0}^{r} a_i \frac{\boldsymbol{x}^i}{\|\boldsymbol{x}^i\|_2}, \tag{3}$$

where $\boldsymbol{x}^i = [x_1^i, x_2^i, \cdots, x_d^i]^\top$ represents element-wise exponentiation, and $\|\cdot\|_2$ denotes the $L_2$ normalization. PolyNorm incorporates normalization operators to rescale different powers into a manageable range, thereby preventing excessively large or small values. This makes the training procedures more stable.

**Integration into Transformer.** The transformer architecture (Vaswani et al., 2017) consists of two alternating modules, Multi-Head Attention (MHA) and position-wise Feed-Forward Networks (FFN). Activation functions predominantly influence the performance of FFN layers. We begin by formalizing the common paradigm of FFN,

$$\text{FFN}_\rho(\boldsymbol{x}) = \rho(\boldsymbol{x}W_1)W_2, \tag{4}$$

where $\rho$ represents the activation function such as ReLU, GELU, PolyReLU, and PolyNorm. We replace the traditional activation function with our proposed PolyCom variants to enhance model capacity and performance, as illustrated in Figure 2.

## 3 THEORETICAL ANALYSIS

From Figure 1, one can see that the expressivity of PolyNorm is greater than or equal to that of PolyReLU. To streamline the analysis, we focus solely on the theoretical properties of PolyReLU, specifically its expressivity and effectiveness. Additional, nonlinear activations such as GELU and SwiGLU can be locally approximated by Taylor polynomials around the origin, which allows us to primarily compare PolyReLU with ReLU and polynomial activations. To avoid confusion, we refer to networks that use ReLU activations as ReLU networks, and those that use PolyReLU activations as PolyReLU networks.

### 3.1 APPROXIMATING RELU NETWORKS BY POLYRELU

In this subsection, we present theoretical results on approximating ReLU networks using PolyReLU networks. The following lemma shows that ReLU, ReLU$^2$, and polynomial activation are special cases of PolyReLU activation, highlighting the superior expressivity of PolyReLU. This implies that PolyReLU has stronger approximation abilities with fewer trainable parameters compared to ReLU and other polynomial activations.

**Lemma 1.** ReLU, ReLU$^2$ *and polynomial activation can be represented by* PolyReLU.

Building on Lemma 1, we can formally prove that any ReLU network can be exactly represented by a PolyReLU network of the same size, as stated in the following theorem.

**Theorem 1.** *Let* $f : [-1, 1]^d \to [-1, 1]$ *be a ReLU network with depth* $L$ *and width* $K$. *Then, there exists a PolyReLU network* $g : [-1, 1]^d \to [-1, 1]$ *of size* $O(LK)$ *such that*

$$f(\boldsymbol{x}) = g(\boldsymbol{x}), \quad \text{for } \forall \boldsymbol{x} \in [-1, 1]^d. \tag{5}$$

This theorem, proved in Appendix A, shows that PolyReLU networks can exactly match the representational power of ReLU networks without increasing the model size.

### 3.2 APPROXIMATING POLYRELU WITH RELU NETWORKS

In this part, we give theoretical results on approximating PolyReLU networks using ReLU networks. The following Lemma 2 demonstrates that the PolyReLU activation can be approximated by a ReLU network within a given error tolerance.

**Lemma 2.** *For the fixed positive integer* $r$ *and the activation* $\text{PolyReLU}(x) = \sum_{i=0}^{r} a_i \text{ReLU}^i(x), x \in [-1, 1]$ *with* $a_i \in [-1, 1]$. *Given any* $\epsilon \in (0, 1)$, *there exists a ReLU network* $f : [-1, 1] \to [-1, 1]$ *with size* $O(\ln^2(1/\epsilon))$, *such that*

$$\max_{x \in [-1, 1]} |f(x) - \text{PolyReLU}(x)| < \epsilon. \tag{6}$$

Lemma 2 establishes an upper bound on the size of a ReLU network needed to approximate a PolyReLU activation function. This result highlights that while ReLU networks can approximate PolyReLU activations, they require a significantly larger number of parameters.

Building on Lemma 2, we derive the following theorem, which provides both upper and lower bounds for approximating PolyReLU networks with ReLU networks.

**Theorem 2.** *Let* $g : [-1, 1]^d \to [-1, 1]$ *be a PolyReLU network with depth* $L$ *and width* $K$, *and PolyReLU activation with order* $r$ *and Lipschitz constant* $\alpha$. *Suppose each neuron computes* $x \mapsto \text{PolyReLU}(a^\top x + b)$ *with the pair* $(a, b)$ *satisfies* $\|a\|_1 + b \leq 1$ *and* $\text{PolyReLU} : [-1, 1] \to [-1, 1]$ *(a, b, and PolyReLU are possibly distinct across neurons). For any given* $\epsilon \in (0, 1)$, *there exists a ReLU network* $f : [-1, 1]^d \to [-1, 1]$ *of size*

$$O\left(LK \ln^2\left(\frac{L\alpha^L}{\epsilon}\right)\right), \tag{7}$$

*such that*

$$\max_{\boldsymbol{x} \in [-1, 1]^d} |f(\boldsymbol{x}) - g(\boldsymbol{x})| < \epsilon. \tag{8}$$

*Conversely, there exists PolyReLU networks cannot be approximated within tolerance* $\epsilon$ *by any ReLU network with a size less than*

$$\Omega\left(KL \ln\left(\frac{1}{\epsilon}\right)\right). \tag{9}$$

Theorem 2 tells us that the total number of trainable parameters required by ReLU networks to approximate a PolyReLU neural network within a tolerance of $\epsilon$ is $O(\ln^2(1/\epsilon))$. Conversely, there exists a PolyReLU network that can not be approximated by ReLU networks of size less than $\Omega(\ln(1/\epsilon))$. Combined with Theorem 1, we conclude that PolyReLU networks are more efficient in terms of representational capacity than ReLU networks.

### 3.3 APPROXIMATION OF GENERAL SMOOTH FUNCTION

Similar to Yarotsky (2017); Boullé et al. (2020), we also explore the universal approximation capabilities of PolyReLU networks in the context of Sobolev spaces (Adams & Fournier, 2003). Specifically, we show that PolyReLU networks achieve the ***optimal approximation rate*** within these spaces, meaning that PolyReLU networks require minimum parameters to approximate general smooth functions in Sobolev spaces, compared with networks with the other activation.

The definition of Sobolev space $\mathcal{W}^{n,\infty}\left([-1,1]^d\right)$ is stated below. The set $[-1,1]^d$ can be replaced by any compact set in $\mathbb{R}^d$, we use it just for the sake of brevity.

**Definition 1** (Sobolev Spaces). *For $n, d \in \mathbb{N}$, Sobolev space $\mathcal{W}^{n,\infty}\left([-1,1]^d\right)$ is defined as*

$$\mathcal{W}^{n,\infty}\left([-1,1]^d\right) = \left\{ f \in L_\infty\left([-1,1]^d\right) \mid \|f\|_{\mathcal{W}^{n,\infty}([-1,1]^d)} < \infty \right\}, \tag{10}$$

*with the norm which is defined as the following*

$$\|f\|_{\mathcal{W}^{n,\infty}([-1,1]^d)} = \max_{\boldsymbol{n}:\|\boldsymbol{n}\|_1 \leq n} \operatorname{ess\,sup}_{\boldsymbol{x}\in[-1,1]^d} \|D^{\boldsymbol{n}} f(\boldsymbol{x})\|_\infty, \tag{11}$$

*where $\boldsymbol{n} \in \mathbb{N}^d$ and $D^{\boldsymbol{n}} f$ is the respective weak derivative of $f$, and* $\operatorname{ess\,sup}$ *means the essential supremum in functional analysis.*

Intuitively, a Sobolev space is a space of functions endowed with a weaker notion of smoothness compared to differentiability and possessing generalized derivatives. The Sobolev space $\mathcal{W}^{n,\infty}\left([-1,1]^d\right)$ contains functions from $C^{n-1}\left([-1,1]^d\right)$ which consists of functions whose derivatives of order $n-1$ are Lipschitz continous. In the sequel, we mainly consider the unit ball within $\mathcal{W}^{n,\infty}\left([-1,1]^d\right)$, which is defined as follows

$$F_{n,d} = \{f \in \mathcal{W}^{n,\infty}\left([-1,1]^d\right) \mid \|f\|_{\mathcal{W}^{n,\infty}([-1,1]^d)} \leq 1\}.$$

With the above definitions established, we can present the following main results. We provide an upper bound on the size of PolyReLU networks required to approximate any function in $F_{n,d}$.

**Theorem 3.** *Suppose that $d, n \in \mathbb{N}$ and $\epsilon \in (0,1)$. For any $f \in F_{d,n}$, there exists a PolyReLU network $g$ with size $O(\epsilon^{-d/n})$ that can approximate $f$ at a given error tolerance $\epsilon$, i.e.,*

$$\max_{\boldsymbol{x}\in[-1,1]^d} \|f(\boldsymbol{x}) - g(\boldsymbol{x})\|_\infty < \epsilon. \tag{12}$$

Theorem 3 indicates that PolyReLU networks can achieve an *optimal approximation rate* of $O(\epsilon^{-d/n})$. In contrast, previous works by Yarotsky (2017) demonstrated that ReLU networks require $O(\epsilon^{-d/n} \ln(1/\epsilon))$ parameters to achieve a similar approximation error. Similarly Boullé et al. (2020) showed that rational neural networks need $O(\epsilon^{-d/n} \ln(\ln(1/\epsilon)))$ parameters for the same task. Therefore, the approximation ability of PolyReLU networks is superior to that of both ReLU networks and rational networks. Furthermore, Theorem 4.2 in DeVore et al. (1989) shows that the total number of parameters required by neural networks to approximate functions in $F_{n,d}$ is $\Omega(\epsilon^{-d/n})$. Therefore, our PolyReLU networks achieve the *optimal approximation rate* in the context of Sobolev spaces. Additional disscution is included in Appendix B.

## 4 EXPERIMENTS

In this section, we demonstrate the expressivity and effectiveness of PolyCom within the transformer through experiments on LLMs.

### 4.1 SETUP

**Baseline.** We evaluate PolyCom across two series of models: a 1B dense model and a Mixture of Experts (MoE) model with 1B active and 7B total parameters. The 1B dense model contains approximately 1.3 billion parameters with an architecture similar to Llama 2 (Touvron et al., 2023).

Table 1: Overall results of the 1B dense model with different activation functions, reported in terms of training loss, validation perplexity, and downstream accuracy (%). ARC-E and ARC-C refer to ARC-Easy and ARC-Challenge, respectively. The best results in each column are highlighted in **bold**. "Avg." denotes the average accuracy of all downstream tasks.

| | Loss↓ | PPL↓ | ARC-E | ARC-C | HellaSwag | PIQA | SciQ | Winograde | Avg.↑ |
|---|---|---|---|---|---|---|---|---|---|
| SwiGLU | 2.19 | 3.22 | 56.61 | 27.47 | 49.23 | 68.61 | 86.10 | **56.83** | 57.47 |
| GELU | 2.20 | 3.24 | 55.43 | 27.73 | 48.42 | 68.12 | 87.40 | 54.78 | 56.98 |
| ReLU | 2.21 | 3.26 | 55.68 | 28.50 | 48.59 | 68.39 | 87.10 | 54.85 | 57.18 |
| PolyReLU | **2.17** | 3.18 | 57.53 | 27.99 | 50.19 | **70.29** | **87.60** | 55.72 | 58.22 |
| PolyNorm | **2.17** | **3.17** | **59.68** | **29.01** | **50.86** | 69.15 | 87.20 | 56.20 | **58.68** |

For the MoE model, we use the OLMoE framework (Muennighoff et al., 2024), which activates 1.3B parameters out of a total of 6.9B parameters. Both models are trained from scratch. We compare the performance of PolyCom with several activation functions, including ReLU, square ReLU, GELU, and SwiGLU. All experiments are conducted on NVIDIA A100-80G GPUs, 32 GPUs for the dense model, and 64 GPUs for the MoE model.

**Model Configuration.** For the dense model, the transformer consists of 24 layers with hidden size $d_{model} = 2048$ and 16 attention heads. In the MoE model, the transformer is composed of 16 layers, with a hidden size of $d_{model} = 2048$, 16 attention heads, and 64 experts. To maintain a consistent number of trainable parameters across all activation functions, we adjust the intermediate size accordingly. Specifically, for SwiGLU, the intermediate size is set to two-thirds that of the other activations in all experiments. More details can be found in Appendix E.

**Datasets.** The dense model is trained on the RedPajama-1T dataset [1] (Computer, 2023), which was developed by the open-source AI community to enable competitive performance against proprietary models. The MoE model is trained on the OLMoE Mix dataset [2] (Muennighoff et al., 2024).

**Hyperparameters.** Unless otherwise specified, we use a 3-order PolyCom by default and initialize the coefficients as $a_i = 1/3$ for $i = 1, 2, 3$ and set $a_0 = 0$. Model weights are randomly initialized. For optimization, we apply the AdamW optimizer with $\beta_1 = 0.9$ and $\beta_2 = 0.95$. All models are trained on sequences of 4096 tokens. For the dense model, we set the initial learning rate to 3e-4, decaying to 1.5e-5 using a cosine scheduler. The MoE model starts with a learning rate of 4e-4, also decaying according to a cosine schedule. We summarize the hyperparameters in Table 7.

**Evaluation.** To evaluate the performance of LLMs with PolyCom, we use a wide range of open benchmarks, including ARC-Easy (Clark et al., 2018), ARC-Challenge (ARC-C) (Clark et al., 2018), HellaSwag (Zellers et al., 2019), PIQA (Bisk et al., 2020), SciQ (Welbl et al., 2017), CoQA (Reddy et al., 2019), Winogrande (Sakaguchi et al., 2021), MMLU (Hendrycks et al., 2021), BoolQ (Clark et al., 2019), COPA (Gordon et al., 2012), CSQA (Talmor et al., 2019), OBQA (Mihaylov et al., 2018), and SocialIQA (Sap et al., 2019). We utilize the LM Eval Harness (Gao et al., 2023) for standardized performance evaluation.

## 4.2 RESULTS ON DENSE MODEL

**Training Dynamics of 1B Dense Model.** Figure 1 compares the training dynamics of the 1B dense model across different activation functions. As shown in the figure, models using PolyReLU and PolyNorm exhibit lower training loss and validation perplexity throughout the training process compared to models utilizing other activation functions. This indicates that PolyCom accelerates the convergence of LLMs. The models with PolyReLU and PolyNorm also consistently outperform others in downstream tasks by large margins, highlighting the advantage of PolyCom in improving the overall expressivity and effectiveness of LLMs.

**Downstream Evaluation.** Table 1 presents the training loss, validation perplexity, and downstream task accuracy (%) after processing 250 billion training tokens. The downstream tasks include ARC-Easy, ARC-Challenge, HellaSwag, PIQA, SciQ, and Winograde. More detailed results are provided

---

[1] RedPajama-1T is available at `https://github.com/togethercomputer/RedPajama-Data`.
[2] OLMoE Mix dataset is available at `https://huggingface.co/datasets/allenai/OLMoE-mix-0924`.

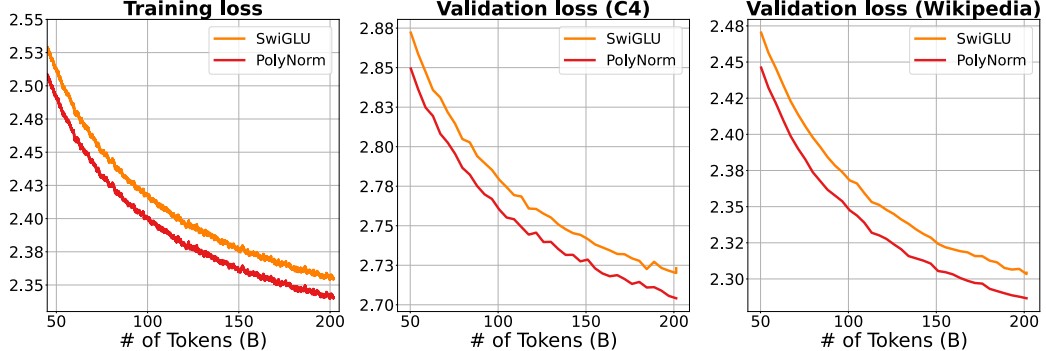

Figure 3: Training and validation loss on C4 and Wikipedia for MoE models with 200 billion training tokens. We compare models using SwiGLU and PolyNorm activation functions. PolyNorm demonstrates lower training and validation losses, indicating faster convergence.

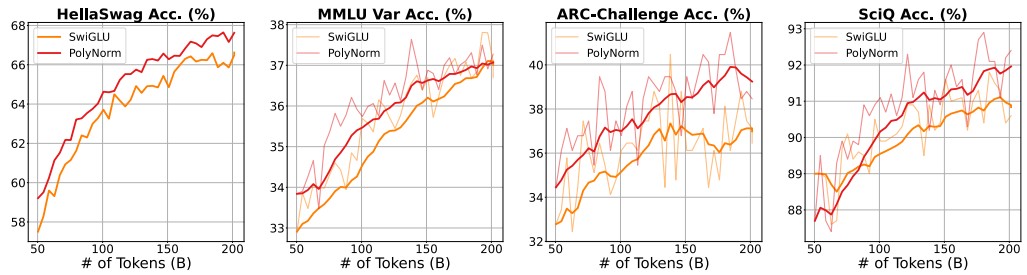

Figure 4: Dynamics of downstream performance on HellaSwag, MMLU Var, ARC-Challenge, and SciQ for MoE models with 200 billion training tokens. Models with PolyNorm significantly outperform those with SwiGLU on downstream tasks.

in Appendix I. The results clearly demonstrate that the PolyCom family (PolyReLU and PolyNorm) outperforms the other activation functions. For instance, PolyNorm outperforms SwiGLU by an average margin of 1.21% across six downstream tasks. This underscores the expressivity and efficiency of PolyCom as an activation function in transformer models.

## 4.3 RESULTS ON MoE MODEL

Our experiments with MoE modes are based on OLMOE-1B-7B, which has 1 billion activate parameters and 7 billion total parameters (Muennighoff et al., 2024). Due to computational constraints, we compare only the PolyNorm activation function, shown to perform best in dense models, with the widely used SwiGLU activation function, which is commonly employed in current LLM architectures.

**Training dynamics of MoE model.** In Figure 3, we report the training and validation loss of MoE models trained on 200 billion tokens. Models using PolyNorm consistently show lower losses compared to those using SwiGLU, indicating that PolyNorm enables faster learning. Figure 4 shows the downstream performance on HellaSwag, MMLU Var[3], ARC-Challenge, and SciQ. PolyNorm outperforms SwiGLU on all tasks, with notable improvements, demonstrating superior generalization capabilities.

**Downstream Evaluation.** Table 2 presents the validation losses on 11 datasets. PolyNorm consistently achieves lower validation losses than SwiGLU across all datasets, with an average improvement of 0.02. In Table 3, we also observe that PolyNorm outperforms SwiGLU on 8 downstream tasks. These results highlight the superior performance of models using the PolyNorm activation function. Additional results can be found in Appendix J.

---

[3]MMLU Var is a variant of MMLU (Hendrycks et al., 2021) using varied few-shots (Muennighoff et al., 2024).

Table 2: Validation losses of MoE models with different activation functions. CC denotes Common Crawl. Best results per column are **bold**.

| Methods | C4 | Books | CC | peS2o | Reddit | Stack | Wiki-pedia | ICE | M2D2 | Pile | Wiki-text | Avg.↓ |
|---|---|---|---|---|---|---|---|---|---|---|---|---|
| SwiGLU | 2.72 | 2.59 | 2.79 | 2.16 | 2.93 | 1.01 | 2.30 | 2.50 | 3.07 | 2.07 | 2.37 | 2.41 |
| PolyNorm | **2.71** | **2.57** | **2.78** | **2.15** | **2.92** | **1.00** | **2.29** | **2.49** | **3.06** | **2.03** | **2.34** | **2.39** |

Table 3: Downstream evaluation results of MoE models with different activation functions. ARC-C, ARC-E, OQA denote ARC-Challenge, ARC-Easy, and OpenbookQA, respectively. Best results per column are in **bold**.

| Tasks | MMLU Var | Hella-Swag | SciQ | ARC-C | ARC-E | PIQA | Wino-Grande | OQA | COPA | Avg.↑ |
|---|---|---|---|---|---|---|---|---|---|---|
| SwiGLU | 37.07 | 66.49 | 90.60 | 37.12 | **71.58** | 76.61 | **62.75** | 39.80 | 83.00 | 62.78 |
| PolyNorm | **37.27** | **67.63** | **92.40** | **38.46** | 70.70 | **77.04** | 62.19 | **40.60** | **84.00** | **63.37** |

## 4.4 ABLATIONS AND ANALYSIS.

**Order of PolyCom.** We first investigate the effect of different orders of PolyCom. We vary the order $r$ of PolyReLU in the range $\{2, 3, 4\}$ and plot the results in Figure 5(a). As seen, the convergence speed improves as the order increases. However, there is no noticeable difference between orders 3 and 4 in terms of convergence speed. Additionally, increasing the order can lead to computational overhead and overflow issues, particularly when using low-precision arithmetic. Based on these observations, we select $r = 3$ as the default order for PolyCom in our experiments, balancing both performance and computational efficiency.

**Different Polynomial Composition Functions.** We evaluate the impact of different polynomial composition functions by comparing PolyReLU, PolyPReLU, PolyNorm, and PolyReLUNorm in Figure 5(b). Our results indicate that PolyNorm, which uses normalization as the composition function, achieves the lowest training loss and best overall performance. This suggests that normalization plays a key role in stabilizing training and enhancing the model's ability to generalize. In contrast, combining ReLU with normalization (PolyReLUNorm) provides intermediate results, suggesting that more complex compositions do not always lead to better outcomes.

**Variants of ReLU.** In Figure 5(c), we compare different variants of the ReLU activation function, including ReLU and ReLU$^2$. PolyReLU consistently outperforms both ReLU and ReLU$^2$ across all tasks, highlighting the benefits of using polynomial composition. This result reinforces the hypothesis that introducing higher-order terms through PolyCom enables the model to capture more complex data interactions, thus improving the expressivity of the activation function without significantly increasing model size or complexity.

**Rank of Weights.** To understand how PolyCom enhances model performance, we analyze the rank of the weights in each FFN layer of the transformer. We use the effective rank (Roy & Vetterli, 2007) to measure the effective dimensionality of weights and its definition is in Appendix E.3. Figure 6 shows that PolyReLU and PolyNorm result in higher weight ranks compared to other activation functions such as SwiGLU, GELU, and ReLU. A higher rank in the weight matrices usually indicates a greater capacity for representing complex patterns in the data. These findings suggest that PolyCom improves the expressibility of transformers by allowing the FFN layers to better utilize their parameters, ultimately leading to better generalization on downstream tasks.

**Layer-wise Similarity.** We further analyze the layer-wise similarity of hidden states using cosine similarity, as illustrated in Figure 7. For both dense and MoE models, we compare SwiGLU with PolyNorm. The results reveal that PolyNorm consistently maintains lower layer-wise similarity compared to SwiGLU, indicating that PolyNorm promotes greater diversity between layers. This diversity likely enables the model to learn more complex representations, as deeper layers are not merely replicating the functionality of earlier ones. Notably, the gap in cosine similarity between PolyNorm and SwiGLU widens in the deeper layers, which are generally more crucial for down-

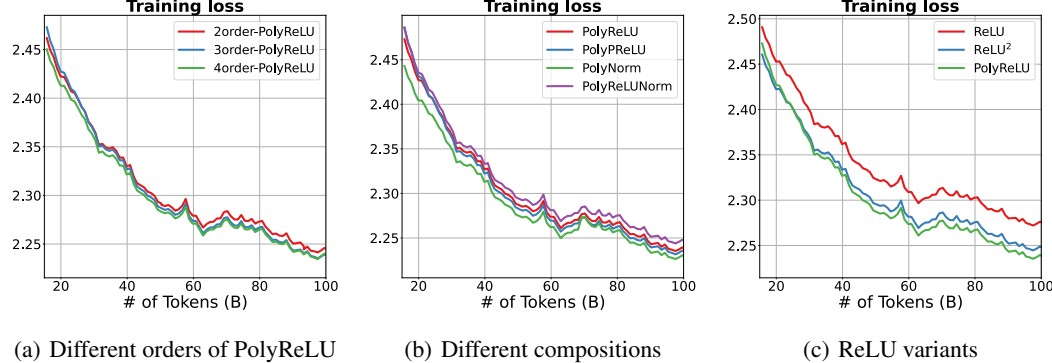

(a) Different orders of PolyReLU     (b) Different compositions     (c) ReLU variants

Figure 5: Training loss for 1B dense models with different activation functions. 5(a): We compare different orders of PolyReLU. 5(b): Comparison of PolyCom with different composition functions. 5(c): Comparison of different variants of ReLU activation function.

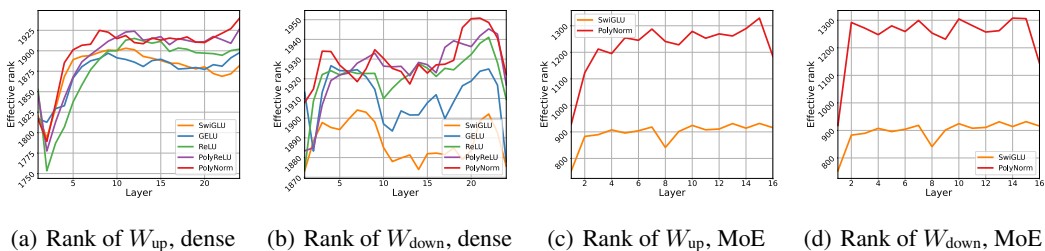

(a) Rank of $W_{\text{up}}$, dense    (b) Rank of $W_{\text{down}}$, dense    (c) Rank of $W_{\text{up}}$, MoE    (d) Rank of $W_{\text{down}}$, MoE

Figure 6: Rank of weights in each FFN. 6(a) & 6(b) for the dense model, 6(c) & 6(d) for the MoE model.

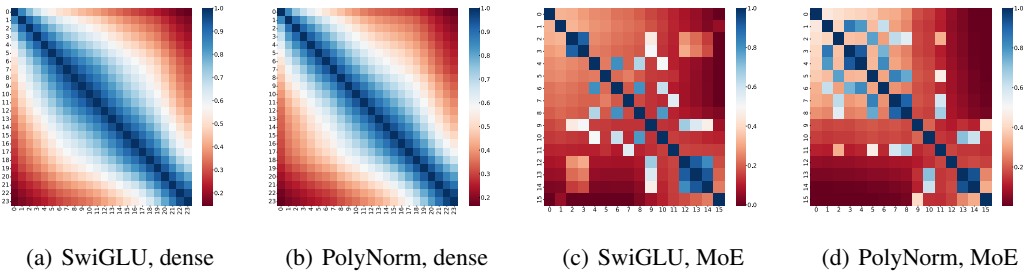

(a) SwiGLU, dense     (b) PolyNorm, dense     (c) SwiGLU, MoE     (d) PolyNorm, MoE

Figure 7: Layer-wise cosine similarity of hidden states. 7(a) &7(b): for 1B dense models with SwiGLU and PolyNorm, respectively. 7(c) & 7(d): for MoE models with SwiGLU and PolyNorm, respectively.

stream task performance. This increased diversity across layers enhances the model's ability to capture complex relationships, thereby improving the overall effectiveness of LLMs.

**Training Stability.** Through extensive experiments, we find both PolyReLU and PolyNorm maintain a stable training process within transformer architectures. Our analysis indicates the combination of normalization operators in transformers and standard gradient clipping strategies effectively stabilizes training dynamics. We specifically design PolyNorm to address potential instability associated with BF16/FP16 precision formats, incorporating normalization operators that rescale powers to a manageable range, thus preventing excessively large or small values. This is particularly beneficial for FP16 training, as demonstrated in Appendix H. In contrast, PolyReLU lacks this normalization feature, which may result in stability issues in non-transformer architectures like ResNet. As shown in Figure 5(a), a 3rd order (our default setting) is sufficient. Based on these findings, we

recommend the following configurations: (1) For transformer-based models, use either PolyNorm or PolyReLU. (2) For non-transformer models or those with lower stability, prefer PolyNorm.

**Computational Overhead and Memory Footprint.** We provide detailed analyses of the runtime and memory overhead for the proposed activation functions, including FLOPs ratios and memory consumption, which are included in Appendix F. Overall, after applying the gradient checkpointing technique, the overhead and memory footprint are acceptable, and there is negligible difference in the training budget required compared to the widely used SwiGLU.

## 5 RELATED WORK

**The design of activation functions** has been a critical area of research in neural networks, directly influencing the performance and capabilities of deep learning models. Early activation functions like Sigmoid and Tanh were widely used due to their smooth nonlinear transformations (Goodfellow et al., 2016). However, these functions faced challenges such as vanishing gradients, making it difficult to train deep networks effectively. The introduction of the Rectified Linear Unit (ReLU) (Nair & Hinton, 2010) mitigated some of these issues by offering a simple, non-saturating nonlinearity, which has since become a standard in many deep learning applications. Variants of ReLU, such as Leaky ReLU (Maas et al., 2013) and Parametric ReLU (PReLU) (He et al., 2015), were developed to address the "dying ReLU" problem by allowing a small, non-zero gradient when the input is negative. Other functions, like the Exponential Linear Unit (ELU) (Clevert, 2015), aimed to provide smoother activation profiles, resulting in better generalization and faster convergence in certain tasks. Moreover, Manessi & Rozza (2018) proposed a combination of weighted base activation functions for further enhancement.

**Polynomial activation functions** (Hornik et al., 1989; Oh et al., 2003), although less commonly used, have been studied in various contexts for their ability to model higher-order, complex relationships more effectively. For instance, Lokhande et al. (2020) introduced Hermite polynomial activations to improve pseudo-label accuracy, while Chrysos et al. (2020) proposed polynomial networks, Π-nets, which apply to various domains such as image and audio processing. Building on this, Chrysos et al. (2023) utilized regularization techniques to enhance the performance of polynomial networks. These works highlight the potential of polynomial functions to increase the expressiveness of neural networks by capturing intricate, higher-order interactions. On the theoretical front, the expressivity and approximation power of polynomial functions have been rigorously explored (Kileel et al., 2019; Kidger & Lyons, 2020; Kubjas et al., 2024). Additionally, (Li et al., 2019) investigated the approximation capabilities of rectified power units (*i.e.,* $\text{ReLU}^2$), demonstrating that they achieve the same approximation rate as PolyReLU.

**The choice of activation function in transformers** has also become an important area of research. Originally developed for natural language processing, transformers (Vaswani et al., 2017) have been effectively adapted for diverse tasks, including image recognition, speech processing, and reinforcement learning. Despite their broad applicability, the activation functions predominantly utilized in transformers, ReLU and GELU, have seen minimal evolution. Recent studies, however, have begun to explore alternatives to these conventional activations. For example, the Swish activation (Ramachandran et al., 2017; Shazeer, 2020) and the Mish activation (Misra, 2019) are smooth and non-monotonic functions that offer potential benefits in model performance and training stability. Additionally, Gated Linear Units (GLU) were proposed by Dauphin et al. (2017), with SwiGLU (Shazeer, 2020), a prominent variant, being used in models such as LLaMA-Series (Touvron et al., 2023).

## 6 CONCLUSIONS

In this paper, we introduce the Polynomial Composition Activation (PolyCom) and demonstrate its effectiveness within transformer models. By enabling the capture of higher-order interactions, PolyCom enhances both the accuracy and convergence rates of these models. Our experiments, conducted across different large language model architectures and multiple benchmarking datasets, confirm that PolyCom consistently outperforms conventional activation functions. Furthermore, ablation studies indicate that PolyCom increases model expressivity by elevating weight rank and reducing redundancy across layers. These findings underscore the significant potential of polynomial-based activations to improve transformer models, thereby paving the way for future research endeavors.

ACKNOWLEDGMENTS

Jinwen Ma was supported by the Natural Science Foundation of China under grant 62071171.

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

# A   OMITTED PROOFS

In this section, we provide the proofs that were omitted in the main body of the paper. The following proofs build upon the work of Yarotsky (2017); Telgarsky (2017); Boullé et al. (2020).

## A.1   PROOF OF LEMMA 1

*Proof of Lemma 1.* For ReLU activation, set $a_1 = 1$, $a_i = 0, \forall i \neq 1$, leading to $\text{PolyReLU}(x) = \text{ReLU}(x)$.

For $\text{ReLU}^2$ activation, set $a_2 = 1$, $a_i = 0, \forall i \neq 2$, giving $\text{PolyReLU}(x) = \text{ReLU}^2(x)$.

For a general polynomial activation, observe that for $\forall x \in \mathbb{R}$ and $i \in \mathbb{N}$:

$$x^i = \text{ReLU}^i(x) + (-1)^i \text{ReLU}^i(-x), \quad \forall x \in \mathbb{R}, \forall i \in \mathbb{N}. \tag{13}$$

Thus, for any polynomial activation of order $r$,

$$\text{Poly}(x) = \text{PolyReLU}_1(x) + \text{PolyReLU}_2(-x), \tag{14}$$

where $\text{PolyReLU}_1(x) = \sum_{i=0}^r a_i \text{ReLU}^i(x)$ and $\text{PolyReLU}_2(x) = \sum_{i=1}^r (-1)^i a_i \text{ReLU}^i(x)$.   □

## A.2   PROOF OF THEOREM 1

The proof is an elementary extension of Lemma 1.

*Proof of Theorem 1.* Using Lemma 1, we can represent the ReLU activation on $\mathbb{R}$ using a PolyReLU activation. Thus, we replace each ReLU activation in the ReLU network $f$ with PolyReLU to construct a new network $g$. Obviously, such $g$ satisfies the above requirements. Hence, the size and structure remain equivalent, and $g$ serves as the PolyReLU network equivalent to the ReLU network.   □

## A.3   PROOF OF LEMMA 2

The proof of Lemma 2 leverages Lemma 3.4 from Telgarsky (2017), which we state below.
**Lemma A.1** (Lemma 3.4 in Telgarsky (2017))**.** *Let $\epsilon \in (0, 1)$ be given. Suppose $p : [0,1]^d \to [-1, 1]$ be a $r$ order polynomial with $s$ monomials and coefficients within $[-1, 1]$. Then there exists a ReLU network $f : [0,1]^d \to [-1, 1]$ of size $O(\min\{sr \ln(sr/\epsilon), sd \ln^2(dsr/\epsilon)\})$ such that $\max_{\boldsymbol{x} \in [0,1]^d} |p(\boldsymbol{x}) - f(\boldsymbol{x})| < \epsilon$.*

Using this result, we now proceed with the proof of Lemma 2.

*Proof of Lemma 2.* First, we observe that $\text{PolyReLU}(x) = \text{Poly}(\text{ReLU}(x))$, where $Poly(x) = \sum_{i=0}^r a_i x^i$ for $x \in [-1, 1]$. By Lemma A.1, there exists a ReLU network $f_1 : [0, 1] \to [-1, 1]$ of size $O(\ln^2(1/\epsilon))$ such that

$$\max_{x \in [0,1]} |f_1(x) - \text{Poly}(x)| < \epsilon. \tag{15}$$

Thus, we construct $f = f_1 \circ \text{ReLU}$ for inputs $x \in [-1, 1]$. This yields that

$$\begin{aligned}
\max_{x \in [-1,1]} |f(x) - \text{PolyReLU}(x)| &= \max_{x \in [-1,1]} |f_1 \circ \text{ReLU}(x) - \text{PolyReLU}(x)| \\
&= \max_{x \in [-1,1]} |f_1(\text{ReLU}(x)) - \text{Poly}(\text{ReLU}(x))| \\
&= \max_{x \in [0,1]} |f_1(x) - \text{Poly}(x)| \\
&< \epsilon.
\end{aligned} \tag{16}$$

Since $f_1$ is a ReLU network, the constructed function $f = f_1 \circ \text{ReLU}$ is also a ReLU network, completing the proof.   □

A.4 PROOF OF THEOREM 2

The lower bound of Theorem 2 follows directly from Theorem 11 in Liang & Srikant (2017), restated here for clarity:

**Lemma A.2** (Theorem 11 in Liang & Srikant (2017)). *Suppose function $f : [0,1]^d \to \mathbb{R}$ is differentiable and strongly convex. Let $\epsilon \in (0,1)$ be given and $\tilde{f}$ be a ReLU network. If $\max_{\boldsymbol{x} \in [0,1]^d} |f(\boldsymbol{x}) - \tilde{f}(\boldsymbol{x})|$, then the network size of $\tilde{f}$ is at least $\Omega(\ln(1/\epsilon))$.*

Lemma A.2 shows that approximating the quadratic function $x^2$ with an error tolerance $\epsilon$ requires a network of size at least $\Omega(\ln(1/\epsilon))$. Since $x^2$ on $[0,1]^d$ is a degradation case of PolyReLU, any ReLU network approximating PolyReLU with error $\epsilon$ must also be at least $\Omega(\ln(1/\epsilon))$ in size. The upper bound is proved in the following.

*Proof of Theorem 2.* Denote $g_i$ as the $i$-th layer of PolyReLU neteeork $f$ for $1 \leq i \leq L$, such that

$$g = g_L \circ g_{L-1} \circ \cdots \circ g_1.$$

For each neuron, since $\|a\|_1 + b \leq 1$, it follows that

$$|a^\top \boldsymbol{x} + b| \leq |a^\top \boldsymbol{x}| + |b| \leq \|a\|_1 \|\boldsymbol{x}\|_\infty + |b| \leq 1, \forall \boldsymbol{x} \in \{\boldsymbol{x}| \|\boldsymbol{x}\|_\infty \leq 1\}. \tag{17}$$

Additionally, note that the range of PolyReLU is $[-1,1]$. Hence, by induction, the output of each neuron remains within $[-1,1]$. For each subnetwork $g_i$, by applying Lemma 2, we can construct a corresponding ReLU network $f_i$ by replacing each PolyReLU activation $p_{i,j}$ in $g_i$ with a ReLU activation. Specifically, for any $i \in [L]^4$ and $j \in [K]$, there exists a ReLU network $f_{i,j} : [-1,1] \to [-1,1]$ that approximates the PolyReLU activation $p_{i,j}$ with given tolerance $\epsilon_i > 0$.

Thus, the network $f_i$ is obtained by replacing each PolyReLU activation $p_{i,j}$ in $g_i$ with its ReLU approximation $f_{i,j}$. Obviously, $f_i$ is a ReLU network whose output dimensions are in the range $[-1,1]$.

Next, we give the approximation error bound. Denote $h_i^g = g_i \circ \cdots \circ g_1$ and $h_i^f = f_i \circ \cdots \circ f_1$ for $i \in [L]$. For the sake of brevity, we assume $h_0^g = h_0^f$ as the identity map in $[-1,1]^d$. Hence, we have $h_i^g = g_i \circ \cdots \circ g_0$ and $h_i^f = f_i \circ \cdots \circ f_0$. Suppose $x \mapsto p_{i,j}(a_{i,j}^\top h_{i-1}^g + b_{i,j})$ be the output of $j$-th neuron of $g_i$. Denote the approximation between the PolyReLU network and the ReLU network at $i$-th layer and $j$-th neuron as $e_{i,j}$. And we use $e_i = \max_{j \in [K]} e_{i,j}$ to denote the approximation error between the PolyReLU network and the ReLU network at $i$-th layer. Then for any $i \in [L]$, we have that

$$\begin{aligned}
e_{i,j} &= \max_{x \in [-1,1]^d} \left| h_{i,j}^g(\boldsymbol{x}) - h_{i,j}^f(\boldsymbol{x}) \right| \\
&= \max_{x \in [-1,1]^d} \left| p_{i,j}(a_{i,j}^\top h_{i-1}^g(\boldsymbol{x}) + b_{i,j}) - f_{i,j}(a_{i,j}^\top h_{i-1}^f(\boldsymbol{x}) + b_{i,j}) \right| \\
&= \max_{x \in [-1,1]^d} \left| p_{i,j}(a_{i,j}^\top h_{i-1}^g(\boldsymbol{x}) + b_{i,j}) - p_{i,j}(a_{i,j}^\top h_{i-1}^f(\boldsymbol{x}) + b_{i,j}) \right. \\
&\qquad \left. + p_{i,j}(a_{i,j}^\top h_{i-1}^f(\boldsymbol{x}) + b_{i,j}) - f_{i,j}(a_{i,j}^\top h_{i-1}^f(\boldsymbol{x}) + b_{i,j}) \right| \\
&\leq \max_{x \in [-1,1]^d} \left| p_{i,j}(a_{i,j}^\top h_{i-1}^g(\boldsymbol{x}) + b_{i,j}) - p_{i,j}(a_{i,j}^\top h_{i-1}^f(\boldsymbol{x}) + b_{i,j}) \right| \tag{18} \\
&\qquad + \max_{x \in [-1,1]^d} \left| p_{i,j}(a_{i,j}^\top h_{i-1}^f(\boldsymbol{x}) + b_{i,j}) - f_{i,j}(a_{i,j}^\top h_{i-1}^f(\boldsymbol{x}) + b_{i,j}) \right| \\
&\leq \max_{x \in [-1,1]^d} \alpha \left| (a_{i,j}^\top h_{i-1}^g(\boldsymbol{x}) + b_{i,j}) - (a_{i,j}^\top h_{i-1}^f(\boldsymbol{x}) + b_{i,j}) \right| + \epsilon_i \\
&\leq \alpha \max_{x \in [-1,1]^d} \|a_{i,j}\|_1 \left\| h_{i-1}^g(\boldsymbol{x}) - h_{i-1}^f(\boldsymbol{x}) \right\|_\infty + \epsilon_i \\
&\leq \alpha \max_{x \in [-1,1]^d} \left\| h_{i-1}^g(\boldsymbol{x}) - h_{i-1}^f(\boldsymbol{x}) \right\|_\infty + \epsilon_i.
\end{aligned}$$

---

[4]We use the notation $[L]$ to denote the set $\{1, 2, \ldots, L\}$.

The first inequality is using the triangular inequality. The second inequality holds because the Lipschitz constant of $p_{i,j}$ is $\alpha$ and the ReLU subnetwork $f_{i,j}$ approximates $p_{i,j}$ with error $\epsilon_i$. In the fourth inequality, we used Hölder's inequality. Since $\|a_{i,j}\|_1 \leq \|a_{i,j}\|_1 + |b_{i,j}| \leq 1$, the fifth inequality holds.

Therefore, we derive the following approximation bound

$$e_i = \max_{j \in [K]} e_{i,j} \leq \alpha \max_{x \in [-1,1]^d} \left\| h_{i-1}^g(\boldsymbol{x}) - h_{i-1}^f(\boldsymbol{x}) \right\|_\infty + \epsilon_i = \alpha e_{i-1} + \epsilon_i, \tag{19}$$

for $\forall i \in [L]$. Since $h_0^g = h_0^f$, we have $e_0 = 0$. Let $\epsilon_i = \epsilon/(L\alpha^{L-i})$ for $\forall i \in [L]$. It follows that

$$e_i \leq \frac{i\epsilon}{L\alpha^{L-i}}, \quad \forall i \in [L]. \tag{20}$$

Hence, the final error at the last layer is bounded by $e_L \leq \epsilon$.

Last, we need to estimate the size of the ReLU network $f$. By Lemma 2, the size of each ReLU subnetwork $f_{i,j}$ is $O(\ln^2(L\alpha^{L-i}/\epsilon))$. Therefore, the total size of the ReLU network $f$ is

$$O\left( \sum_{i=1}^L K \ln^2 \left( \frac{L\alpha^{L-i}}{\epsilon} \right) \right) = O\left( KL \ln^2 \left( \frac{L\alpha^L}{\epsilon} \right) \right), \tag{21}$$

where we use the fact that

$$\sum_{i=1}^L \ln^2 \left( \frac{L\alpha^{L-i}}{\epsilon} \right) = \sum_{i=1}^L \left( \ln \left( \frac{L\alpha^L}{\epsilon} \right) - i \ln \alpha \right)^2 = O\left( L \ln^2 \left( \frac{L\alpha^L}{\epsilon} \right) \right). \tag{22}$$

This completes the proof. $\qquad \square$

### A.5 Proof of Theorem 3

Before proving Theorem 3, we begin by introducing a few useful lemmas.

**Lemma A.3** (Proposition 1 in Yarotsky (2017)). *Let $M \in \mathbb{N}$ and $\rho : \mathbb{R} \to \mathbb{R}$ be any continuous piece-wise linear function with $M$ breakpoints. Then the following two statements hold:*

- *For a network with activation $\rho$, depth $L$ and width $K$, there exists a ReLU network with the same depth $L$ and width $O(MK)$ that computes the same function as the original network.*

- *Conversely, if a ReLU network has depth $L$ and width $K$, there exists a network with activation $\rho$, depth $L$ and width $K$ that computes the same function on a bounded input domain $\mathcal{D}$.*

This result, Combined with Lemma 1, directly leads to the following corollary, which demonstrates that PolyReLU networks can represent any piece-wise linear function exactly on $\mathbb{R}$.

**Corollary A.1.** *Let $M \in \mathbb{N}$ and $\rho : \mathbb{R} \to \mathbb{R}$ be any continuous piece-wise linear function with $M$ breakpoints. Then there exists a PolyReLU network $g$ of size $O(M)$ such that*

$$\rho(x) = g(x), \quad \forall x \in \mathbb{R}.$$

In a similar manner to Proposition 10 in Boullé et al. (2020), we can show that PolyReLU networks can represent powers $x^n$ exactly for any $n \in \mathbb{N}$.

**Lemma A.4.** *Suppose $n, r \in \mathbb{N}$ and $r \geq 2$. Then $x^n$ can be represented exactly by a PolyReLU network $g$ with an $r$-th order PolyReLU activation and size $O(\ln^2(n))$.*

*Proof of Lemma A.4.* We first prove that $x^n$ can be represented exactly by a polynomial network $\hat{g}$ with $r$-th order polynomial activation and having size $O(\ln^2(n))$. Based on $\hat{g}$, we construct a PolyReLU network $g$ that satisfies the requirements.

By expressing $n$ in base $r$, we have that

$$x^n = \prod_{i=0}^k x^{c_i r^i} = \prod_{i=0}^k \left( x^{c_i} (x^r)^i \right), \tag{23}$$

where $k = \lfloor \log_r n \rfloor$, $n = \sum_{i=0}^{k} c_i r^i$, and $c_i \in \{0, 1, 2, \ldots, r-1\}$. Each $x^{c_i r^i}$ can be represented by a polynomial network with $i+1$ layers and width 1. It follows that $x^n$ can be represented by a polynomial network of size

$$\sum_{i=0}^{k} (i+1) = O(k^2) = O(\ln^2(n)). \tag{24}$$

By Lemma 1, we know that a PolyReLU activation can represent a polynomial activation. Hence, there exists a PolyReLU network $g$ with an $r$-th order activation and size $O(\ln^2(n))$ such that

$$g(x) = x^n, \quad \forall x \in \mathbb{R}.$$

$\square$

With the above lemmas, we can now prove Theorem 3.

*Proof of Theorem 3.* The proof is composed of two parts. We first approximate $f$ by local Taylor polynomials and continuous piece-wise linear functions and then represent these functions using PolyReLU networks, following Yarotsky (2017); Boullé et al. (2020).

**Part 1.** Suppose $N$ is a positive integer. We begin by dividing $[-1, 1]^d$ into a grid of $(2N+1)^d$ functions:

$$\sum_{\boldsymbol{m}} \phi_{\boldsymbol{m}}(\boldsymbol{x}) = 1, \quad \phi_{\boldsymbol{m}}(\boldsymbol{x}) = \prod_{i=1}^{d} \varphi \left( 3N \left( x_k - \frac{m_k}{N} \right) \right), \quad \forall \boldsymbol{x} = (x_1, x_2, \ldots, x_d) \in [-1, 1]^d,$$

where $\boldsymbol{m} = (m_1, m_2, \ldots, m_d) \in \{-N, -(N-1), \ldots, 0, \ldots, N\}^d$, and $\varphi$ is defined as

$$\varphi(x) = \begin{cases} 1, & |x| < 1, \\ 0, & 2 < |x|, \\ 2 - |x|, & 1 \le |x| \le 2. \end{cases}$$

This function has the following properties:

$$\max_{x \in \mathbb{R}} |\varphi(x)| = 1, \quad \max_{\boldsymbol{x} \in [-1,1]^d} \|\phi_{\boldsymbol{m}}(\boldsymbol{x})\|_\infty = 1, \tag{25}$$

$$\text{supp } \phi_{\boldsymbol{m}} = \left\{ \boldsymbol{x} \,\middle|\, \left\| \boldsymbol{x} - \frac{\boldsymbol{m}}{N} \right\|_\infty < \frac{2}{3N} \right\}, \forall \boldsymbol{m} \in \{-N, -(N-1), \ldots, N\}^d. \tag{26}$$

**Part 2.** We use a degree-$(n-1)$ local Taylor approximation of the function $f$, defined as

$$f_N(\boldsymbol{x}) = \sum_{\boldsymbol{m} \in \{-N, \ldots, N\}^d} \phi_{\boldsymbol{m}}(\boldsymbol{x}) P_{\boldsymbol{m}}(\boldsymbol{x}), \tag{27}$$

where $P_{\boldsymbol{m}}$ is the degree-$(n-1)$ Taylor polynomial of $f$ at $\boldsymbol{x} = \boldsymbol{m}/N$, i.e.,

$$P_{\boldsymbol{m}}(\boldsymbol{x}) = \sum_{\boldsymbol{n}:\|\boldsymbol{n}\|_1 < n} \frac{1}{\boldsymbol{n}!} D^{\boldsymbol{n}} f \left( \frac{\boldsymbol{m}}{N} \right) \left( \boldsymbol{x} - \frac{\boldsymbol{m}}{N} \right)^{\boldsymbol{n}}, \tag{28}$$

with conventions $\boldsymbol{n}! = \prod_{i=1}^{d} n_i!$ and $\left( \boldsymbol{x} - \frac{\boldsymbol{m}}{N} \right)^{\boldsymbol{n}} = \prod_{i=1}^{d} \left( x_i - \frac{m_i}{N} \right)^{n_i}$.

The approximation error between $f$ and $f_N$ can be bounded as follows

$$\begin{aligned} |f(\boldsymbol{x}) - f_N(\boldsymbol{x})| &= \left| \sum_{\boldsymbol{m} \in \{-N, \ldots, N\}^d} \phi_{\boldsymbol{m}} \left( f(\boldsymbol{x}) - P_{\boldsymbol{m}}(\boldsymbol{x}) \right) \right| \\ &\le \sum_{\boldsymbol{m}:\|x-\frac{\boldsymbol{m}}{N}\|_\infty < \frac{2}{3N}} |f(\boldsymbol{x}) - P_{\boldsymbol{m}}(\boldsymbol{x})| \\ &\le 2^d \max_{\boldsymbol{m}:\|x-\frac{\boldsymbol{m}}{N}\|_\infty < \frac{2}{3N}} |f(\boldsymbol{x}) - P_{\boldsymbol{m}}(\boldsymbol{x})| \\ &\le \frac{2^d}{n!} \left( \frac{2d}{3N} \right)^n \max_{\boldsymbol{n}:\|\boldsymbol{n}\|_1 = n} \operatorname*{ess\,sup}_{\boldsymbol{x} \in [-1,1]^d} \|D^{\boldsymbol{n}} f(\boldsymbol{x})\|_\infty \\ &\le \frac{2^d}{n!} \left( \frac{2d}{3N} \right)^n. \end{aligned} \tag{29}$$

The first inequality is because of the triangular inequality and Eq. (25). In the second inequality, we used the fact that $\forall x \in [-1,1]^d$ belongs to the support of at most $2^d$ functions $\phi_{\boldsymbol{m}}$. The third inequality is a bound for the Taylor remainder and the fourth inequality uses the definition of $F_{n,d}$. Let

$$N = \left\lfloor \frac{2d}{3} \left( \frac{2^d}{n!\epsilon} \right)^{\frac{1}{n}} \right\rfloor + 1, \tag{30}$$

we have that

$$\max_{\boldsymbol{x} \in [-1,1]^d} |f(\boldsymbol{x}) - f_N(\boldsymbol{x})| < \epsilon. \tag{31}$$

Next, we construct a PolyReLU network $g_N$ to represent $f_N$ exactly. Let $a_{\boldsymbol{m},\boldsymbol{n}} = \frac{1}{\boldsymbol{n}!} D^{\boldsymbol{n}} f\left( \frac{\boldsymbol{m}}{N} \right)$. Since $\|f\|_{\mathcal{W}^{n,\infty}([-1,1]^d)} \le 1$, $|a_{\boldsymbol{m},\boldsymbol{n}}| \le 1$ for any $\boldsymbol{m}, \boldsymbol{n}$, we rewrite $f_N$ as

$$f_N(\boldsymbol{x}) = \sum_{\boldsymbol{m} \in \{-N,\dots,N\}^d} \sum_{\boldsymbol{n}:\|\boldsymbol{n}\|_1 < n} a_{\boldsymbol{m},\boldsymbol{n}} \phi_{\boldsymbol{m}}(\boldsymbol{x}) \left( \boldsymbol{x} - \frac{\boldsymbol{m}}{N} \right)^{\boldsymbol{n}}. \tag{32}$$

Therefore, $f_N$ is composed of at most $d^n(2N+1)^d$ functions $\phi_{\boldsymbol{m}}(\boldsymbol{x}) \left( \boldsymbol{x} - \frac{\boldsymbol{m}}{N} \right)^{\boldsymbol{n}}$. Since $\phi_{\boldsymbol{m}}(\boldsymbol{x}) = \prod_{i=1}^{d} \varphi\left( 3N\left( x_k - \frac{m_k}{N} \right) \right)$ and each $\varphi\left( 3N\left( x_k - \frac{m_k}{N} \right) \right)$ is a continuous piece-wise linear function, we can apply Corollary A.1, which guarantees that there exists a PolyReLU network $\hat{\phi}_{\boldsymbol{m}}$ of size $O(d)$ that can exactly represent $\phi_{\boldsymbol{m}}$ on $\mathbb{R}^d$, i.e., $\hat{\phi}_{\boldsymbol{m}}(\boldsymbol{x}) = \phi_{\boldsymbol{m}}(\boldsymbol{x}), \forall \boldsymbol{x} \in \mathbb{R}^d$. For $\left( \boldsymbol{x} - \frac{\boldsymbol{m}}{N} \right)^{\boldsymbol{n}} = \prod_{i=1}^{d} \left( x_i - \frac{m_i}{N} \right)^{n_i}$, by Lemma A.4, we know that there exists a PolyReLU network $g_{\boldsymbol{m}}$ of size at most $O(d \ln^2(n))$ such that $g_{\boldsymbol{m}}(\boldsymbol{x}) = \left( \boldsymbol{x} - \frac{\boldsymbol{m}}{N} \right)^{\boldsymbol{n}}, \forall \boldsymbol{x} \in \mathbb{R}^d$. Combining these results, we can now construct a larger PolyReLU network $g_n$ as follows

$$g_N(\boldsymbol{x}) = \sum_{\boldsymbol{m} \in \{-N,\dots,N\}^d} \sum_{\boldsymbol{n}:\|\boldsymbol{n}\|_1 < n} a_{\boldsymbol{m},\boldsymbol{n}} \hat{\phi}_{\boldsymbol{m}}(\boldsymbol{x}) g_{\boldsymbol{m}}(\boldsymbol{x}), \tag{33}$$

where the total size of the network is

$$O\left( d^n(2N+1)^d(d + d\ln^2(n)) \right) = O(\epsilon^{-\frac{d}{n}}).$$

Here, we use Eq. (30) to determine the size bound in terms of the error tolerance $\epsilon$. Clearly, we have

$$f_N(\boldsymbol{x}) = g_N(\boldsymbol{x}), \quad \forall \boldsymbol{x} \in \mathbb{R}^d. \tag{34}$$

Hence, we conclude that

$$\max_{\boldsymbol{x} \in [-1,1]^d} |f(\boldsymbol{x}) - g_N(\boldsymbol{x})| = \max_{\boldsymbol{x} \in [-1,1]^d} |f(\boldsymbol{x}) - f_N(\boldsymbol{x})| < \epsilon. \tag{35}$$

This completes the proof. □

## B DISCUSSION OF THE OPTIMAL APPROXIMATION RATE

For convenience, we state Theorem 4.2 in DeVore et al. (1989) in the following.

**Theorem 4** (Theorem 4.2 in DeVore et al. (1989)). *Let $\mathcal{X}$ be a Banach space $L_q$ on $\mathbb{R}^d$, $1 \le q \le \infty$. If $F_{n,d}^p = \{f \in \mathcal{X} | \|f\|_{\mathcal{W}^{n,p}} \le 1\}, 1 \le p \le q, n \in \mathbb{N}$, then*

$$\sup_{f \in F_{n,d}^p} \inf_{\theta \in \mathbb{R}^m} \|f - \mathcal{M}(\theta)\|_q \ge Cm^{-\frac{n}{d}}, \tag{36}$$

*where $\mathcal{M}$ be a mapping from $\mathbb{R}^m$ into $\mathcal{X}$ which associate with each $\theta \in \mathbb{R}^m$ the element $\mathcal{M}(\theta) \in \mathcal{X}$, and $C$ is a constant.*

Particularly, let $q = p = \infty$ and $\mathcal{X} = L_\infty[-1,-1]^d$, the above theorem tells us that the approximation error of the neural networks with $m$ parameters to approximate $F_{n,d}^\infty$, *i.e.*, $F_{d,n}$, is larger than $Cm^{-\frac{n}{d}}$. Therefore, given error tolerance $\epsilon$, we have

$$\epsilon \ge Cm^{-\frac{n}{d}}. \tag{37}$$

It follows that

$$m \ge C^{\frac{d}{n}} \epsilon^{-\frac{d}{n}}. \tag{38}$$

Hence, the total number of parameters required by neural networks to approximate functions in $F_{n,d}$ is $\Omega(\epsilon^{-\frac{d}{n}})$. Combining with Theorem 3, we have that our PolyReLU networks achieve the optimal approximation rate in the context of Sobolev spaces.

## C  ACTIVATION FUNCTIONS

We provide definitions of several commonly used non-linear activation functions in Table 4.

Table 4: Definition of activation functions.

| Activation | Definition |
|---|---|
| ReLU (Nair & Hinton, 2010) | $\text{ReLU}(x) = \max\{x, 0\}$ |
| ReLU$^2$ (So et al., 2021) | $\text{ReLU}^2(x) = \max\{x, 0\}^2$ |
| ReLU6 (Krizhevsky et al., 2010) | $\text{ReLU6}(x) = \min(\max\{x, 0\}, 6)$ |
| Leaky ReLU (Maas et al., 2013) | $\text{LeakyReLU}(x) = \begin{cases} x, & \text{if } x \geq 0 \\ ax, & \text{otherwise} \end{cases}$, $a \in (0, 1)$ is a constant |
| RReLU (Xu et al., 2015) | $\text{RReLU}(x) = \begin{cases} x & \text{if } x \geq 0 \\ ax & \text{otherwise,} \end{cases}$

$a$ is randomly sampled from uniform distribution |
| Parametric ReLU (PReLU)

(He et al., 2015) | $\text{PReLU}(x) = \begin{cases} x, & \text{if } x \geq 0 \\ ax, & \text{otherwise ,} \end{cases}$

$a$ is a learnable parameter |
| Tanh | $\text{Tanh}(x) = \frac{\exp(x) - \exp(-x)}{\exp(x) + \exp(-x)}$ |
| Softplus (Glorot et al., 2011) | $\text{Softplus}(x) = \frac{1}{a} * \log(1 + \exp(ax))$,

$a$ is a constant (default 1.0) |
| Mish (Misra, 2019) | $\text{Mish}(x) = x * \text{Tanh}(\text{Softplus}(x))$ |
| Sigmoid | $\text{Sigmoid}(x) = \sigma(x) = \frac{1}{1 + \exp(-x)}$ |
| SiLU(Swish) (Ramachandran et al., 2017) | $\text{SiLU}(x) = x * \sigma(x)$ |
| ELU (Clevert, 2015) | $\text{ELU}(x) = \begin{cases} x, & \text{if } x > 0 \\ a * (\exp(x) - 1), & \text{if } x \leq 0, \end{cases}$

$a$ is a constant (default 1.0) |
| CELU (Barron, 2017) | $\text{CELU}(x) = \max(0, x) + \min(0, \alpha * (\exp(x/a) - 1))$,

$a$ is a constant (default 1.0) |
| GELU (Hendrycks & Gimpel, 2016) | $\text{GELU}(x) = x * \Phi(x)$,

$\Phi(x)$ is CDF for Gaussian distribution |
| GLU (Dauphin et al., 2017) | $\text{GLU}(x) = \sigma(xW) \otimes (xV)$ |
| SwiGLU (Shazeer, 2020) | $\text{SwiGLU}(x) = \text{SiLU}(xW) \otimes (xV)$,

$W, V$ are learnable parameters |
| Poly | $\text{Poly}(x) = \sum_{i=0}^{r} a_i x^i$, $a_i, i \in [r]$ are learnable parameters |

# D PyTorch Implementation of PolyCom

PyTorch implementations of PolyReLU and PolyNorm are provided in the following.

---

**Algorithm 1** PyTorch-Style Implementation of PolyReLU

---

```python
import torch
from torch.utils.checkpoint import checkpoint
import torch.nn.functional as F

def _poly(x, weight, bias, order=3):
    return sum(weight[i] * (x ** (i+1)) for i in range(order)) + bias

class PolyReLU(torch.nn.Module):
    def __init__(self):
        super(PolyReLU, self).__init__()
        self.weight = torch.nn.Parameter(torch.ones(3) / 3)
        self.bias = torch.nn.Parameter(torch.zeros(1))

    def forward(self, x, checkpointing=True):
        x = F.relu(x)
        if checkpointing:
            return checkpoint(_poly, x, self.weight, self.bias, use_reentrant=False)
        return _poly(x, self.weight, self.bias)
```

---

---

**Algorithm 2** PyTorch-Style Implementation of PolyNorm

---

```python
def _norm(x, eps=1e-6):
    return x * torch.rsqrt(x.pow(2).mean(-1, keepdim=True) + eps)

def _poly_norm(x, weight, bias, order=3):
    return sum(weight[i] * _norm(x ** (i+1)) for i in range(order)) + bias

class PolyNorm(torch.nn.Module):
    def __init__(self):
        super(PolyNorm, self).__init__()
        self.weight = torch.nn.Parameter(torch.ones(3) / 3)
        self.bias = torch.nn.Parameter(torch.zeros(1))

    def forward(self, x, checkpointing=True):
        if checkpointing:
            return checkpoint(_poly_norm, x, self.weight, self.bias, use_reentrant=False)
        return _poly_norm(x, self.weight, self.bias)
```

---

# E Experimental Details

## E.1 Architecture

Table 5 outlines the model architecture used for the 1B dense model. To ensure comparable numbers of training parameters across different activation functions, we adjust the intermediate sizes accordingly. For SwiGLU, the intermediate size is set to 5504, while for other activation functions, it is set to 8256.

Table 6 outlines the model architecture used for the MoE models. Similarly, the intermediate size for SwiGLU is set to 1024, while for other activation functions, it is set to 1536.

Table 5: Model architecture of the 1B dense model.

| Params | Hidden Size | Context Length | Intermediate Size | Attention Heads | Hidden Layers |
|--------|-------------|----------------|-------------------|-----------------|---------------|
| 1.3B | 2048 | 4096 | 5504/8256 | 16 | 24 |

## E.2 Hyperparameters

In Table 7, we list the hyperparameters that we use by default at training time for all our experiments for the 1B dense model and MoE-1B-7B, unless stated otherwise.

Table 6: Model architecture of MoE model.

| Activate Params | Total Params | Hidden Size | Intermediate Size | Attention Heads |
|---|---|---|---|---|
| 1.3B | 6.9B | 2048 | 1024/1536 | 16 |
| Hidden Layers | Exports | Active Exports | Context Length | Weight Tying |
| 16 | 64 | 8 | 4096 | no |

Table 7: Pretraining hyperparameters for the 1B dense model and MoE-1B-7B.

|  | 1B dense model | MoE-1B-7B |
|---|---|---|
| Optimizer | AdamW | AdamW |
| Learning Rate (LR) | 3E-4 | 4E-4 |
| Minimum LR | 3E-5 | 5E-5 |
| LR Schedule | cosine | cosine |
| Weight Decay | 0.1 | 0.1 |
| $\beta_1$ | 0.9 | 0.9 |
| $\beta_2$ | 0.95 | 0.95 |
| Gradient Clipping | 1 | 1 |
| Warmup Tokens | 620,000,000 | - |
| Warmup Steps | - | 2000 |
| Init Distribution | normal | trunc normal |
| Init std | $1/\sqrt{2.5d}$ | $1/\sqrt{2.5d}$ |
| Init Truncation | - | $3\times$ std |
| Load Balancing Loss Weight | - | 0.01 |
| Router z-loss Weight | - | 0.001 |

### E.3 DEFINITION OF EFFECTIVE RANK

We adopt the concept of effective rank from Roy & Vetterli (2007) to measure the effective dimensionality of a matrix. Given a matrix $A$ with Singular Value Decomposition (SVD) $A = U\Sigma V^\top$, where $\Sigma$ is a diagonal matrix containing singular values $\sigma_1 \geq \sigma_2 \geq \cdots \geq \sigma_n \geq 0$. we define the singular value distribution as $p_i = \sigma_i / \sum_{j=0}^{n} \sigma_j, i \in [n]$. The effective rank of $A$ is then given by

$$\text{Erank}(A) = \exp\left(-\sum_{i=0}^{n} p_i \ln p_i\right). \tag{39}$$

## F COMPUTATIONAL COMPLEXITY ANALYSIS

For the sake of simplicity, we only calculate the computational complexity in one-layer Feed-Forward Networks (FFN) since activation only. Support input tensor of FFN is $x \in \mathbb{R}^{B \times S \times H}$, where $B$, $L$, and $H$ are the batch size, length of the sequence, and hidden size, respectively. Roughly, the relationship between computational FLOPs and model parameters can be regarded as proportional [5]. Therefore, we can estimate the proportion of the computational cost incurred by the activation function calculations within the total computational cost of the FFN matrix computations ($24BSH^2$). The FLOPs ratio is calculated as

$$\text{FLOPs ratio} = \frac{\text{FLOPs for activation}}{24BSH^2}.$$

It is important to note that the overhead and proportion often vary for different model sizes, so we provide the corresponding formulas directly and take $H = 1024$, $B = 4$ (each device), $S = 4096$, using BF16 precision as an example. For PolyReLU and PolyNorm, we use the 3-order default setting. The results are summarized in the following tables:

---

[5] https://blog.eleuther.ai/transformer-math/

Table 8: Comparison of computational complexity for different activation functions **without gradient checkpointing**.

| Method | Intermediate Size | FLOPs for activation | FLOPs ratio | Memory Overhead |
|--------|-------------------|----------------------|-------------|-----------------|
| ReLU | $4H$ | $4BSH$ | $\frac{1}{6H} = 0.016\%$ | $4BSH = 128MB$ |
| GELU | $4H$ | $72BSH$ | $\frac{3}{H} = 0.29\%$ | $10BSH = 320MB$ |
| SwiGLU | $\frac{8}{3}H$ | $\frac{112}{3}BSH$ | $\frac{14}{9H} = 0.15\%$ | $8BSH = 256MB$ |
| ReLU$^2$ | $4H$ | $8BSH$ | $\frac{1}{3H} = 0.032\%$ | $8BSH = 256MB$ |
| PolyNorm | $4H$ | $72BSH$ | $\frac{3}{H} = 0.29\%$ | $12BSH = 384MB$ |
| PolyReLU | $4H$ | $40BSH$ | $\frac{5}{3H} = 0.16\%$ | $8BSH = 256MB$ |

Table 9: Comparison of computational complexity for different activation functions **with gradient checkpointing**.

| Method | Intermediate Size | FLOPs for activation | FLOPs ratio | Memory Overhead |
|--------|-------------------|----------------------|-------------|-----------------|
| ReLU | $4H$ | $8BSH$ | $\frac{1}{3H} = 0.033\%$ | $0$ |
| GELU | $4H$ | $144BSH$ | $\frac{6}{H} = 0.59\%$ | $0$ |
| SwiGLU | $\frac{8}{3}H$ | $\frac{224}{3}BSH$ | $\frac{28}{9H} = 0.30\%$ | $0$ |
| ReLU$^2$ | $4H$ | $16BSH$ | $\frac{2}{3H} = 0.065\%$ | $0$ |
| PolyNorm | $4H$ | $144BSH$ | $\frac{6}{H} = 0.59\%$ | $0$ |
| PolyReLU | $4H$ | $80BSH$ | $\frac{10}{3H} = 0.33\%$ | $0$ |

We assume that the scale of the input tensor is set to $[-1, 1]$. In this case, the FLOPs for both tanh and exp are approximately 10 each. For a fair comparison, the intermediate size of models with SwiGLU activations is set to 8/3H to keep the overall numbers of parameters constant.

In practice, we utilized gradient checkpointing [6] to reduce the additional memory overhead to 0. While this may introduce a certain computational overhead, given the overall modest computational cost of the activation functions, the overall increase in GPU memory and computational cost is quite small.

## G  SCALING CURVES

In Figure 8, we present the training loss scaling curves for dense models utilizing the activation functions SwiGLU, PolyReLU, and PolyNorm. As illustrated in the figure, both PolyReLU and PolyNorm consistently outperform SwiGLU across model sizes ranging from 110M to 1.3B parameters.

The model sizes used for the scaling law experiments are detailed in Table 10, and all models employ the hyperparameters specified for 1B dense models, as listed in Table 7. Models with 110M, 226M, and 502M parameters were trained on 200B tokens.

Table 10: Model sizes for scaling laws experiments.

| Params | Hidden Size | Context Length | Intermediate Size | Attention Heads | Hidden Layers |
|--------|-------------|----------------|-------------------|-----------------|---------------|
| 110M | 768 | 2048 | 2048/3072 | 16 | 12 |
| 226M | 1024 | 2048 | 2560/3840 | 16 | 16 |
| 502M | 1536 | 2048 | 4096/6144 | 16 | 16 |
| 1.3B | 2048 | 4096 | 5504/8256 | 16 | 24 |

---

[6]https://pytorch.org/docs/stable/checkpoint.html

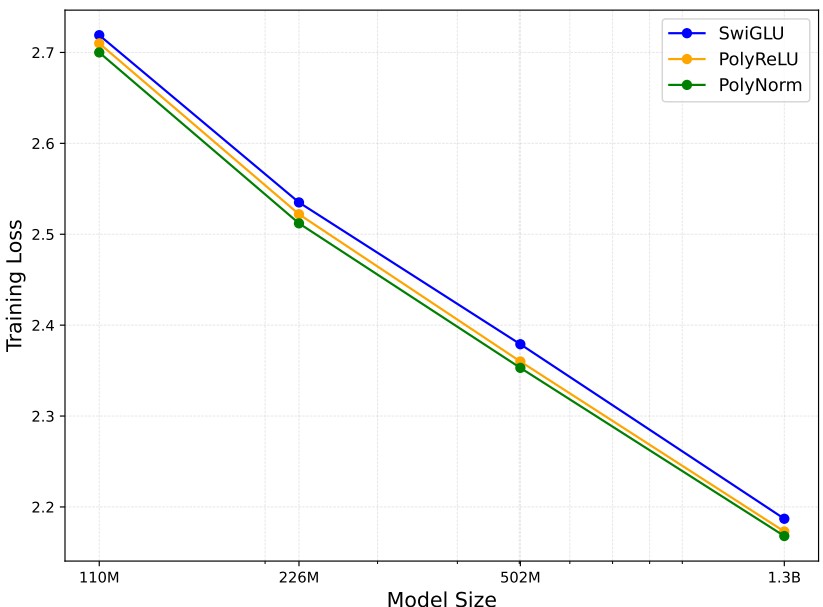

Figure 8: Scaling curves of models with different activation functions.

## H    EXPERIMENTS ON VISION

To evaluate the effectiveness of PolyCom beyond language modeling, we trained ResNet50 (He et al., 2016) on ImageNet-1K (Deng et al., 2009) following the settings of timm (Wightman, 2019). For comparison, we replaced the ReLU activation in ResNet50 with PolyNorm and reported the training loss and top-1/top-5 accuracy on the evaluation set, as shown in Figure 9. The results demonstrate that PolyNorm outperforms ReLU by a significant margin in terms of training loss, top-1 accuracy, and top-5 accuracy. Specifically, PolyNorm achieves a lower training loss of **2.026** compared to ReLU's 2.121, and improves top-1 and top-5 accuracy to **75.117%** and **92.099%**, surpassing ReLU by +0.204 and +0.068, respectively.

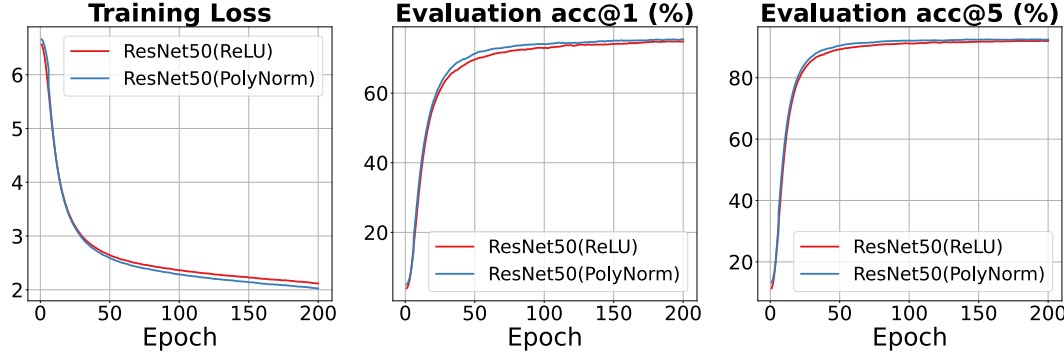

Figure 9: Training loss and evaluation accuracy on ImageNet-1K for ResNet50 models with ReLU and PolyNorm activations. PolyNorm achieves lower training loss and higher top-1/top-5 accuracy, demonstrating improved performance.

# I  ADDITIONAL RESULTS ON DENSE MODEL

More detailed results from our ablation studies are shown in Figures 10, 11, and 12. These figures illustrate the training loss, validation loss, and validation perplexity (PPL) for the 1B dense model under different configurations.

The results of the 1B dense models trained on 400 billion tokens are presented in Figure 13. As shown in the figure, models employing PolyReLU and PolyNorm consistently achieve significantly better performance compared to SwiGLU.

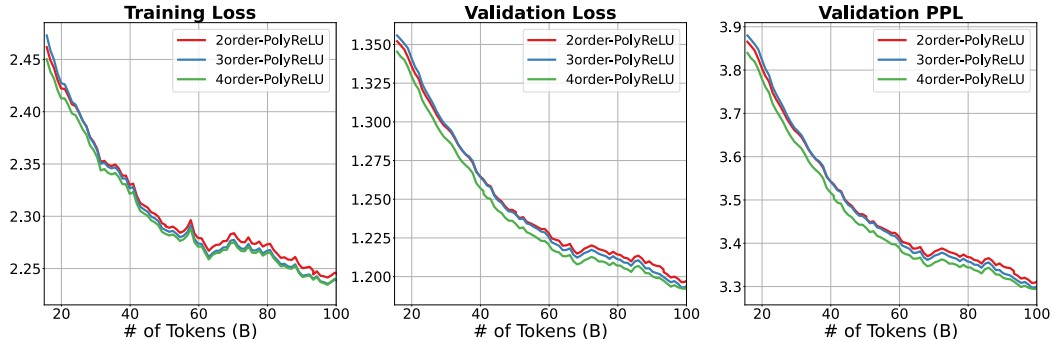

Figure 10: training loss, validation loss, and validation perplexity (PPL) for the 1B dense model with different orders of PolyReLU activation functions.

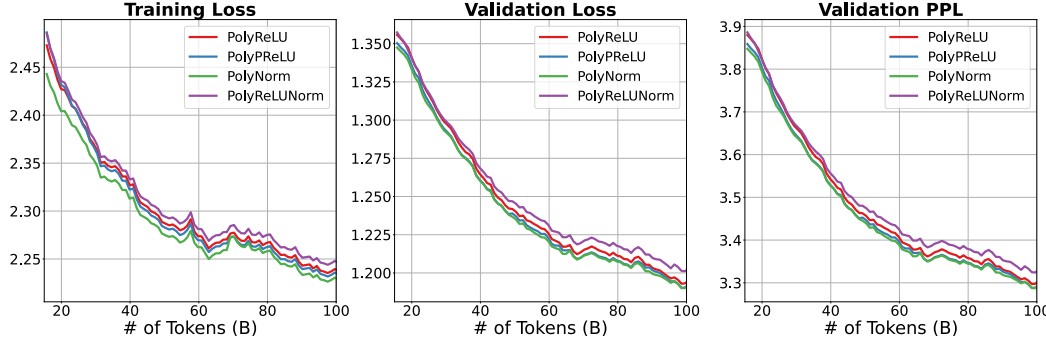

Figure 11: training loss, validation loss, and validation perplexity (PPL) for the 1B dense model with different polynomial compositions.

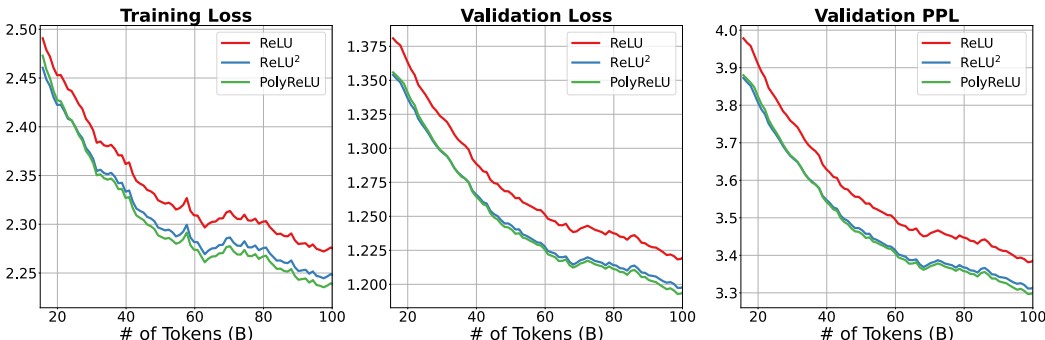

Figure 12: Training loss, validation loss, and validation perplexity (PPL) for the 1B dense model with different variants of ReLU activation functions.

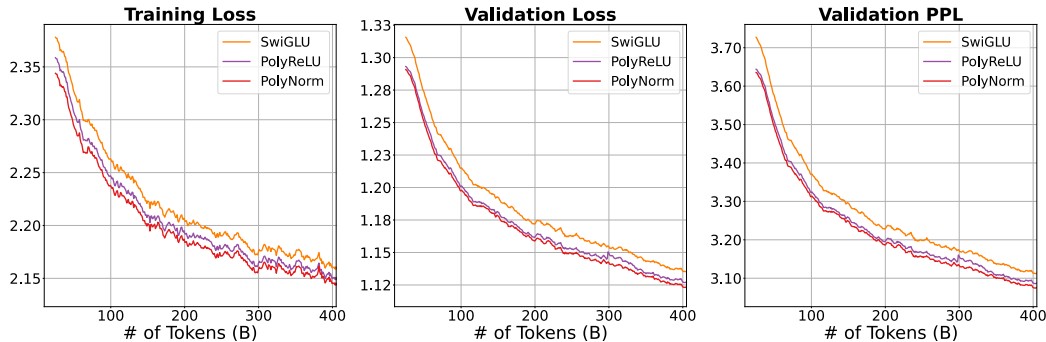

Figure 13: Training loss, validation loss, and validation perplexity (PPL) for the 1B dense model with 400 billion training tokens.

## J  ADDITIONAL RESULTS ON MOE MODEL

More results for the MoE model are provided in Figure 14, showcasing validation losses and downstream evaluations after 200 billion training tokens. The comparison highlights models with different activation functions, such as SwiGLU and PolyNorm. As shown, models with PolyNorm exhibit lower training and validation losses, along with superior downstream performance.

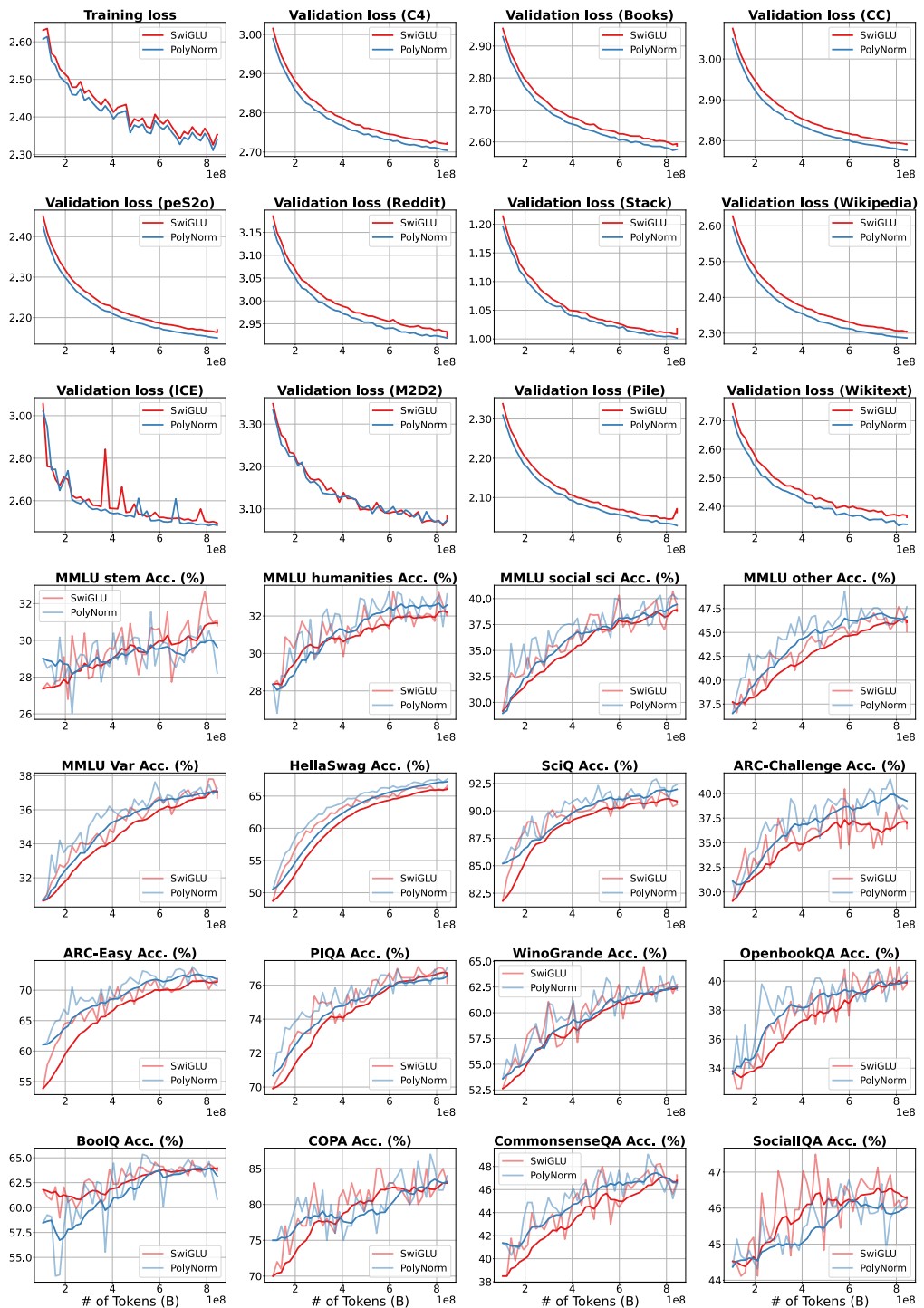

Figure 14: Validation loss and downstream evaluations for MoE models with 200 billion training tokens, comparing SwiGLU and PolyNorm activation functions. PolyNorm shows superior performance in terms of lower loss and better downstream results.

