# OpenReview forum: "Polynomial Composition Activations: Unleashing the Dynamics of Large Language Models"
_ICLR.cc/2025/Conference — ICLR 2025 Poster_

### Official Review · Reviewer_TiU9 · 2024-11-03

**Soundness:** 3
**Presentation:** 4
**Contribution:** 3
**Rating:** 6
**Confidence:** 4

**Summary:**

This paper introduces PolyCom, a polynomial composition activation for transformers. Through theoritical analysis, PolyCom is shown to enhance expressivity and effectiveness over other activations. Empirical experiments on large language models (LLMs), both dense and sparse, demonstrate that replacing standard activations with PolyCom enables LLMs to capture higher-order data interactions, improving accuracy and convergence rates.

**Strengths:**

1) The authors demonstrate the expressivity of the proposed activation function both theoretically and empirically, using effective rank analysis and layer-wise similarity metrics. These methods provide insights into how the activation enhances the model’s ability to represent complex patterns and distinctions between layers, showcasing its potential advantages over traditional activation functions.

2) Extensive experiments on downstream tasks, along with improved convergence rates in the learning curves with fixed parameter size model, highlight the potential of the proposed activation function.

3) Overall, the paper is clearly written, with the description of polynomial activation easy to follow, and the results presented concisely.

**Weaknesses:**

1) Computational complexity analysis is not provided. Can the authors provide some analysis on the inference time throughput of the proposed activations with others?

2) How do the authors address stability or manage potential exploding gradient issues with the polynomial activation?

Typos: Line 274: Dowmstream Evaluation -> Downstream Evaluation

**Questions:**

Check the weaknesses.

---

> ### Author Response · Authors · 2024-11-20
> **Response to Reviewer TiU9**
>
> Thank you for your thoughtful comments and time. We have tried our best to address your concerns and revised our paper accordingly.
>
> ---
>
> Q1. Computational complexity analysis is not provided.
>
> A1. We analyze the runtime overhead introduced by the activation functions using a typical feedforward network (FFN) with input tensor $x\in \mathbb{R}^{B \times S \times H}$, where $B$，$S$ and $H$ represent the batch size, sequence length, and hidden size, respectively. The relationship between computational FLOPs and model parameters can generally be regarded as proportional (as discussed in [Eleuther AI’s transformer math](https://blog.eleuther.ai/transformer-math/)). Below, we estimate the proportion of the computational cost incurred by activation function calculations within the total computational cost of the FFN matrix computations ($24BSH^2$). The FLOPs ratio is calculated as:
>
> $$\text{FLOPs ratio} = \frac{\text{FLOPs for activation}}{24BSH^2}$$
>
> The results are summarized in the following table:
>
> | Method | ReLU | GeLU | SwiGLU | ReLU^2 | 3rd-order PolyNorm | 3rd-order PolyReLU |
> | --- | --- | --- | --- | --- | --- | --- |
> | Intermediate Size | 4H | 4H | 8/3H | 4H | 4H | 4H |
> | FLOPs for activation | 4BSH |72BSH |112/3BSH |8BSH |72BSH |40BSH|
> | FLOPs ratio (H=1024) | 1/(6H)=0.016% | 3/H=0.29% | 14/(9H)=0.15% | 1/(3H)=0.032% | 3/H=0.29% | 5/(3H)=0.16% |
>
> Note:
> - We assume that the scale of the input tensor is set to [-1, 1]. In this case, the FLOPs for both tanh and exp are approximately 10 each.
> - To reduce memory overhead during large language model pretraining, we typically employ gradient checkpointing (refer to [PyTorch Docs](https://pytorch.org/docs/stable/checkpoint.html)). Although this approach incurs some additional computational cost, its overall impact on GPU memory and runtime is minimal.
>
> ---
>
> Q2. How do the authors address stability or manage potential exploding gradient issues with the polynomial activation?
>
> A2. Stability or potential exploding gradient issues were not observed in our training process. The normalization operators in the transformer block effectively stabilized the training, as evidenced by the consistent trends shown in Figure 1 and Figure 3 in the paper. Additionally, for PolyNorm, we hypothesize that its integrated normalization mechanisms contribute further to stabilizing the training process. These results suggest that the activation functions operate reliably within the proposed framework without requiring additional stability interventions.
>
> ---
>
> Thank you for highlighting the typos; we have corrected them accordingly.

---

> > ### Author Response · Authors · 2024-11-27
> > **Follow-Up on Submission Review**
> >
> > Dear Reviewer TiU9,
> >
> > We sincerely appreciate the time and effort you’ve put into reviewing our submission. Your feedback has been instrumental in refining and enhancing the quality of our work.
> >
> > As the deadline for submitting a revised version of the PDF (November 27, AoE) is approaching, we wanted to follow up and ask if there are any remaining questions or concerns we can address. We're more than happy to provide any additional information or clarification you might need.
> >
> > Regarding the computational complexity and runtime overhead, we conducted a thorough analysis in the "[**Global Response to the Computational Overhead and Memory Footprint**](https://openreview.net/forum?id=CbpWPbYHuv&noteId=oIwh3l1V1D)" comment. The code corresponding to this analysis has also been included. The results indicate that the introduced overhead and memory footprint are acceptable, and there is no significant difference in the cost between our proposed method and the existing structures. However, our approach achieve a 1.5x acceleration in convergence. Thus, we believe that our activation functions can make some valuable contributions to the open-source community.
> >
> > Once again, we deeply appreciate your time and effort, and we would be truly grateful if you could re-evaluate the paper's rating.
> >
> > Best regards

---

> > ### Comment · Reviewer_TiU9 · 2024-11-27
> >
> > I appreciate the authors' efforts in addressing the computational complexity and taking the time to answer my questions. I have a couple of follow-up questions:
> >
> > The authors mentioned they did not encounter any instability issues. Is this primarily attributed to the normalization inherent in the transformer architecture? Does this observation also hold for the ResNet50 experiments? Since the authors provided additional ResNet50 results in response to another reviewer's question, could you clarify whether PolyNorm or PolyReLU was used in those experiments? Additionally, does the order of BatchNorm and activations influence the stability in this context?

---

> ### Author Response · Authors · 2024-11-28
>
> Thank you for the follow-up questions about training stability, which is indeed an important point. We appreciate the opportunity to further elaborate on this matter.
>
>   **Inherent Reasons for Stability:**
>  Currently, the reasons why PolyReLU and PolyNorm can maintain a stable training process within the transformer architecture lack a definitive theoretical explanation. A plausible hypothesis is that the normalization operators within transformers, combined with commonly used gradient clipping strategies, effectively stabilize the training process. And we will explore this topic in future research.
>
>   **Key Experimental Observations:**
>   From extensive experiments, we observed that:
>   1. Both PolyReLU and PolyNorm exhibit stable behavior in **transformer** architectures.
>   2. In **ResNet-50** training, while PolyNorm remains stable, PolyReLU encounters stability issues.
>   For ResNet-50 experiments, we simply replaced ReLU with PolyNorm while preserving the original sequence of BatchNorm and activation.
>
> **Recommendations for Practical Usage:**
>   Actually, when designing these activation functions, we have taken into account the widespread adoption of **BF16/FP16** precisions in contemporary model training procedures. To mitigate potential instability, we specifically introduced PolyNorm, which includes normalization operators to rescale different powers to a manageable range, thereby avoiding excessively large or small values. It is particularly beneficial for **FP16** training, as in ResNet-50. PolyReLU doesn't include this normalization property, which may introduce some potential stability issues in other non-transformer-based architectures such as ResNet.
>
>   Also, our experiments reveal that for orders ranging from 2 to 4, the choice of order has minimal impact on stability. Indeed, extremely high orders may pose inherent stability risks. As shown in Figure 5(a), the utilization of 3-order (our default setting) is sufficient.
>
>   Based on these findings, we recommend the following settings:
>   1. For **transformer-based** models, use **PolyNorm** or **PolyReLU** .
>   2. For **non-transformer-based** or less stable models, use **PolyNorm**.
>
>   As highlighted in the title and abstract of our paper, in the current stage, our focus is on the proposed activation functions and their compatibility with **LLMs** or **transformer** architectures. Investigating their effectiveness in other model structures remains an important future research direction.

---

> ### Author Response · Authors · 2024-12-02
>
> Dear Reviewer TiU9,
>
> As the review deadline (**December 2nd**) approaches, we would like to extend our sincere gratitude for the time and effort you have invested in reviewing our work and considering our responses. We are following up to ensure that our replies have fully addressed your concerns and to ask if there is any further clarification or information we can provide.

---

### Official Review · Reviewer_NvaU · 2024-11-04

**Soundness:** 3
**Presentation:** 4
**Contribution:** 3
**Rating:** 8
**Confidence:** 3

**Summary:**

- The authors propose PolyCom, a set of polynomial composition activations that are tailored especially to the needs of transformer models. Theoretical and empirical evaluations compare the expressivity and other capabilities for PolyCom to common pre-existing activations, including traditional ReLU and SwiGLU.
PolyCom applies a composition of a polynomial function and another function $\rho$. Two variants of - PolyCom are focused on. Type 1 PolyCom applies $\rho$ before the polynomial exponent, while Type 2 applies the polynomial exponent before $\rho$ (Equation 1 in the paper). The paper’s evaluations concentrate on one specific substantiation for each type:
  - For Type 1, the authors use PolyReLU. This activation sets $\rho$ to ReLU and applies $\rho$ before the polynomial.
  - For Type 2, the authors use PolyNorm. This activation has the L2 normalization for $\rho$ and applies the polynomial before $\rho$.
- The theory results focus on PolyReLU and proceed in three steps:
  - In section 3.1, the authors show that the sets of all ReLU and ReLU^2 networks are subsets of the set of all PolyReLU networks, indicating that PolyReLU has stronger approximation abilities than ReLU or ReLU^2
  - The main result of Section 3.2 (Theorem 2) is that the size of any ReLU network that approximates a PolyReLU network with depth L and width K within tolerance $\epsilon$ must be at least $\Omega(KL \ln(\epsilon^{-1}))$. By this, authors conclude that PolyReLU networks are more efficient than ReLU networks in terms of representational capacity.
  - Theoretical results build to Section 3.3, which shows in Theorem 3 that PolyReLU networks achieve the optimal approximation rate: that is, there exists a PolyReLU network of size $O(\epsilon^{-d/n})$ that can approximate arbitrary function $f$ in a unit ball within a Sobolev space $F_{d,n}$.
- Empirical evaluations in section 4 apply PolyCom to one dense model with 1B parameters and one MoE model with 1B active and 7B total parameters. Comparison activations are ReLU, ReLU^2, GELU, and SwiGLU. Training datasets are RedPajama-1T for the dense and OLMoE Mix for the MoE model, and several other datasets are considered for downstream fine-tuning tasks. Authors state that lower training/validation loss and downstream task accuracy for PolyCom activations are indicative that PolyCom accelerates the convergence of LLMs and increases model expressivity, with PolyNorm performing generally better than PolyReLU as well. Ablations include the polynomial order of PolyCom and the choice of polynomial composition function $\rho$, and the rank of model weights and layer-wise similarity are also compared favorably towards PolyNorm.

**Strengths:**

- The idea of activation functions with better expressivity, without adding more trainable parameters to the model, is significant, useful and interesting
- The construction/definition of PolyCom is very flexible and serves as a good base for future experimentation with other activation variants beyond the specific instantiations of PolyReLU and PolyNorm
- The combination of both theoretical and empirical results is stronger than including just either type of result alone
- Theoretical results are approachable and elegant
- Paper is well-structured and well-written: it’s easy to follow and there were no confusing typos or other writing issues
- Empirical evaluations are conducted on several different downstream datasets, adding strength to the authors’ claims

**Weaknesses:**

- No discussion of the differences in computational needs between PolyCom and prior activations: does it slow down training time a lot, or do you need much more memory? On lines 416-417 where you’re comparing different orders of PolyCom, you mention higher orders leading to overhead and overflow; I would want to hear more about that
- Empirical results do not seem to be averaged across runs; I don’t see it stated anywhere. Results tables and graphs contain no error bars. Especially because the empirical results are fairly small in many areas, such as in Table 1, I would appreciate more evidence that these patterns hold across training seeds/different random initializations at the start of training
- Other types of polynomial activations, such as those cited in Section 5’s paragraph on polynomial activation functions, are not compared against in the empirical results. Seemingly, no explanation is given for leaving these functions out.
- Small comment: more discussion of the optimal approximation rate or a formal statement of DeVore et al.’s theorem 4.2 (cited on line 275) inside of your paper (even the appendix) could be nice for readers

**Questions:**

- Would you mind including more results on the computational requirements required for PolyCom versus other activations, as I mention in the limitations section? It could also be interesting to include an experiment where PolyCom is compared to other activations, and model size/parameter count is somehow adjusted to ensure that models using PolyCom and models using other activations have comparable computational requirements
- It’s stated that PolyReLU and PolyNorm have equivalent expressivity, e.g. on lines 164-165. Do you have a proof for this?
- Have you tried any PolyCom activation in a non-transformer model? What causes you to think that they’re so good for transformers in particular, as opposed to also for other (deep-learning) architectures?

---

> ### Author Response · Authors · 2024-11-20
> **Response to Reviewer NvaU (1/2)**
>
> We would like to thank the reviewer for your time and constructive comments. We address your concerns below.
>
> ---
>
> Q1. Computational complexity analysis, memory, and overflow issue for higher orders of PolyCom.
>
> A1.  We analyze the runtime overhead introduced by the activation functions using a typical feedforward network (FFN) with input tensor $x\in \mathbb{R}^{B \times S \times H}$, where $B$，$S$ and $H$ represent the batch size, sequence length, and hidden size, respectively. The relationship between computational FLOPs and model parameters can generally be regarded as proportional (as discussed in [Eleuther AI’s transformer math](https://blog.eleuther.ai/transformer-math/)). Below, we estimate the proportion of the computational cost incurred by activation function calculations within the total computational cost of the FFN matrix computations ($24BSH^2$). The FLOPs ratio is calculated as:
>
> $$\text{FLOPs ratio} = \frac{\text{FLOPs for activation}}{24BSH^2}$$
>
> The results are summarized in the following table:
>
> | Method | ReLU | GeLU | SwiGLU | ReLU^2 | 3rd-order PolyNorm | 3rd-order PolyReLU |
> | --- | --- | --- | --- | --- | --- | --- |
> | Intermediate Size | 4H | 4H | 8/3H | 4H | 4H | 4H |
> | FLOPs for activation | 4BSH |72BSH |112/3BSH |8BSH |72BSH |40BSH|
> | FLOPs ratio (H=1024) | 1/(6H)=0.016% | 3/H=0.29% | 14/(9H)=0.15% | 1/(3H)=0.032% | 3/H=0.29% | 5/(3H)=0.16% |
>
> Note:
> - We assume that the scale of the input tensor is set to [-1, 1]. In this case, the FLOPs for both tanh and exp are approximately 10 each.
> - To reduce memory overhead during large language model pretraining, we typically employ gradient checkpointing (refer to [PyTorch Docs](https://pytorch.org/docs/stable/checkpoint.html)). Although this approach incurs some additional computational cost, its overall impact on GPU memory and runtime is minimal.
>
> Overflow issues in higher-order polynomials can be particularly problematic in the context of BF16. Due to the limited dynamic range of BF16, operations involving polynomials of higher degrees can lead to intermediate values exceeding this range, causing numerical overflow.
>
> ---
>
> Q2. Empirical results do not seem to be averaged across runs.
>
> A2. This is an excellent point. Historically, in the early stages of deep learning development, smaller-scale models (e.g., ResNet50 with 26M parameters, ELMo with 94M parameters) were more sensitive to random seeds, as they could significantly influence the optimization trajectory. However, with the advent of large-scale models and abundant training resources, the training process has become remarkably stable, provided that parameter initialization follows a consistent distribution.
>
> For instance, our MoE model contains 6.9B parameters and was trained on 200B tokens using 64 A100-SXM-80GB GPUs, with a batch size of 400M tokens over a span of 7 days. This large-scale training ensures a stable gradient optimization direction, effectively nullifying the impact of random seed variability. As a result, most recent advancements in large-scale model architectures do not account for seed variation.
>
> For downstream metrics, we employed a greedy decoding strategy, ensuring stable and reproducible results. Acknowledging the inherent fluctuations during training, we have included metric evolution plots in Figures 1, 4, and 11, which clearly demonstrate the consistent and stable performance improvements achieved by our proposed method. This evidence should alleviate concerns regarding potential variability in our results.
>
> ---
>
> Q3. Other types of polynomial activations.
>
> A3. The resource-intensive nature of experiments (each requires at least 32 A100 GPUs for several days),  making it hard to include all activation variants. We have carefully selected the most representative activation functions, focusing on the variants that have demonstrated superior performance. Specifically, we have included PolyReLU due to its simplicity and PolyNorm for its effectiveness.
>
> In Section 5, we have conducted experiments with other activation functions for 100 billion tokens and compared training losses across them. Within the domain of large language models (LLM), the training loss exhibits a strong correlation with downstream task metrics, where a lower loss is indicative of better downstream performance. This relationship is further substantiated by the validation loss and perplexity (PPL) trends, as illustrated in Figures 8 through 10 in Appendix F.

---

> ### Author Response · Authors · 2024-11-20
> **Response to Reviewer NvaU (2/2)**
>
> Q4. More discussion of the optimal approximation rate or a formal statement of DeVore et al.’s theorem 4.2.
>
> A4. For clarity, we restate Theorem 4.2 from [1] as follows.
>
> **Theorem 4.2** [1]**.** Let $\mathcal{X}$ be a Banach space $L_q$ on $\mathbb{R}^d$, $1\leq q\leq \infty$. If $F_{n,d}^p=\{f \in \mathcal{X} | \|f\|_{\mathcal{W}^{n,p}} \leq 1\}, 1\leq p\leq q, n \in \mathbb{N}$ , then
>
> $$\sup_{f\in F_{n.d}^p} \inf_{\theta \in \mathbb{R}^m} \|f-\mathcal{M}(\theta)\|_q\geq C m^{-\frac{n}{d}},$$
>
> where $\mathcal{M}$ be a mapping from $\mathbb{R}^m$ into $\mathcal{X}$ which associate with each $\theta \in \mathbb{R}^m$ the element $\mathcal{M}(\theta) \in \mathcal{X}$, and $C$ is a constant.
>
> Particularly, let $q=p=\infty$ and $\mathcal{X}= L_{\infty}[-1,-1]^d$, the theorem indicates that the approximation error for neural networks with $m$ parameters approximating $F_{n,d}^{\infty}$, i.e.,  $F_{n,d}$, is bounded below by $C m^{-\frac{n}{d}}$. Given an error tolerance $\epsilon$, we have
>
> $$
>  \epsilon \geq C m^{-\frac{n}{d}}.$$
>
> which implies:
>
> $$m \geq C^{\frac{d}{n}} \epsilon^{-\frac{d}{n}}.$$
>
> Thus, the total number of parameters required by a neural network to approximate functions in $F_{n,d}$ is at least $\Omega(\epsilon^{-\frac{d}{n}})$. Combined with Theorem 3 in our paper, we establish that PolyReLU networks achieve the optimal approximation rate in the context of Sobolev spaces.
>
> [1].Ronald A DeVore, Ralph Howard, and Charles Micchelli. Optimal nonlinear approximation. Manuscripta mathematica.
>
> ---
>
> Q5. Computational complexity analysis.
>
> A5.  Please refer to Answer A1 of Q1 for a detailed discussion on computational complexity analysis.
>
> ---
>
> Q6. It’s stated that PolyReLU and PolyNorm have equivalent expressivity, e.g. on lines 164-165. Do you have a proof for this?
>
> A6. Thank you for pointing out the less precise expression. We have rephrased the sentence as follows: "From Figure 1, one can see that the expressivity of PolyNorm is greater than or equal to that of PolyReLU."
>
> The claim is primarily supported through the empirical evidence provided in the paper. As can be observed in Figure 1, Figure 6 and Figure 7, both PolyReLU and PolyNorm exhibit superior expressivity in comparison to other activation functions, with PolyNorm demonstrating equal or greater expressive capacity than PolyReLU.
>
> ---
>
> Q7.  Resuts for non-transformer models.
>
> A7.  To evaluate the effectiveness of PolyCom beyond transformer models, we conducted experiments using ResNet50 on ImageNet, following the settings provided by timm [1]. In these experiments, we replaced the ReLU activation in ResNet50 with PolyCom and recorded the training loss, top-1, and top-5 accuracy on the evaluation set. The results are summarized in the tables below.
>
> | Training Loss | 50 epoch | 100 epoch | 150 epoch | 200 epoch |
> | --- | --- | --- | --- | --- |
> | ResNet50(ReLU) | 2.586 | 2.342 | 2.203 | 2.121 |
> | ResNet50(PolyCom) | **2.531**(-0.055) | **2.259**(-0.083) | **2.117**(-0.086) | **2.026**(-0.095) |
>
> | Evaluation acc@1/acc@5 | 50 epoch | 100 epoch | 150 epoch | 200 epoch |
> | --- | --- | --- | --- | --- |
> | ResNet50(ReLU) | 70.089/89.510 | 72.971/91.108 | 74.197/91.736 | 74.913/92.031 |
> | ResNet50(PolyCom) | **71.502/90.294**(+1.413/+0.784) | **73.530/91.581**(+0.559/+0.473) | **74.685/91.978**(+0.488/+0.242) | **75.117/92.099**(+0.204/+0.068) |
>
> These results demonstrate that PolyCom consistently outperforms ReLU in terms of both training loss and evaluation accuracy. The improvements in acc@1 and acc@5 become smaller as the training progresses, which we attribute to the **inherent** **overfitting** tendency of ResNet50 on ImageNet.
>
> References:
>
> [1]. Ross Wightman. PyTorch Image Models. https://github.com/rwightman/pytorch-image-models

---

> > ### Comment · Reviewer_NvaU · 2024-11-26
> >
> > Thank you very much for these clarifying explanations and willingness to update the draft-- with these changes, I will increase my score to 8.

---

> > > ### Author Response · Authors · 2024-11-27
> > > **Appreciation for Your Feedback and Updated Evaluation**
> > >
> > > We are delighted that the revisions align with your expectations, and we are grateful for your updated evaluation. Thank you once again for your engagement and encouragement. Your constructive suggestions are really helpful in enhancing the quality of our paper.

---

### Official Review · Reviewer_NayG · 2024-11-04

**Soundness:** 3
**Presentation:** 3
**Contribution:** 3
**Rating:** 8
**Confidence:** 4

**Summary:**

This paper introduces polynomial composition activations (PolyCom), which is shown to be theoretically more expressive relative to common activation functions like ReLU and has an optimal approximation rate for general smooth functions in Sobolev spaces. Experiments show PolyCom, especially PolyNorm, achieves significantly better performance per token for training 1B parameters dense language models and 7B total parameters MoE model compared to SwiGLU, GeLU, and other ReLU variants.

**Strengths:**

- PolyReLU and PolyNorm show nontrivial performance gains for training language models with >1B parameters, even compared to strong baselines such as SwiGLU and squared ReLU.
- The experiments are comprehensive, covering both pre-training and downstream evaluations.
- PolyReLU has strong theoretical guarantees, showing it's more expressive than ReLU and has an optimal approximation rate for general smooth functions in Sobolev spaces.
- The paper is well-presented.

**Weaknesses:**

- The paper does not discuss potential overhead in using PolyCom. For example, naively, using PolyCom would increase the activation memory by a factor of $r$ (set to 3 in the experiments) compared to ReLU.
- The analysis section is not completely convincing. Why is a higher effective rank better? Both GeLU and SwiGLU have lower effective ranks than ReLU, but they achieve better performance.
- It's not clear that PolyNorm MoE has lower layer-wise cosine similarity in Figure 7 compared to SwiGLU MoE.
- An important unaddressed question is how the benefit of PolyCom scales to larger models. For example, is it most significant for smaller models and vanishing for larger models? I suspect this is not the case, but showing some evidence would be important.

**Questions:**

- What is the memory and runtime overhead of switching from, e.g., squared ReLU to PolyCom?
- Can you show how PolyCom affects the scaling laws of loss v.s. model size or training computed by, for example, training additional smaller models?

---

> ### Author Response · Authors · 2024-11-20
> **Response to Reviewer NayG (1/2)**
>
> Thanks for your careful reading and critical review. We address your concerns below and hope you find them satisfactory.
>
> ---
>
> Q1. What is the memory and runtime overhead of switching from, e.g., squared ReLU to PolyCom?
>
> A1: We analyze the runtime overhead introduced by the activation functions using a typical feedforward network (FFN) with input tensor $x\in \mathbb{R}^{B \times S \times H}$, where $B$，$S$ and $H$ represent the batch size, sequence length, and hidden size, respectively. The relationship between computational FLOPs and model parameters can generally be regarded as proportional (as discussed in [Eleuther AI’s transformer math](https://blog.eleuther.ai/transformer-math/)). Below, we estimate the proportion of the computational cost incurred by activation function calculations within the total computational cost of the FFN matrix computations ($24BSH^2$). The FLOPs ratio is calculated as:
>
> $$\text{FLOPs ratio} = \frac{\text{FLOPs for activation}}{24BSH^2}$$
>
> The results are summarized in the following table:
>
> | Method | ReLU | GeLU | SwiGLU | ReLU^2 | 3rd-order PolyNorm | 3rd-order PolyReLU |
> | --- | --- | --- | --- | --- | --- | --- |
> | Intermediate Size | 4H | 4H | 8/3H | 4H | 4H | 4H |
> | FLOPs for activation | 4BSH |72BSH |112/3BSH |8BSH |72BSH |40BSH|
> | FLOPs ratio (H=1024) | 1/(6H)=0.016% | 3/H=0.29% | 14/(9H)=0.15% | 1/(3H)=0.032% | 3/H=0.29% | 5/(3H)=0.16% |
>
> Note:
> - We assume that the scale of the input tensor is set to [-1, 1]. In this case, the FLOPs for both tanh and exp are approximately 10 each.
> - To reduce memory overhead during large language model pretraining, we typically employ gradient checkpointing (refer to [PyTorch Docs](https://pytorch.org/docs/stable/checkpoint.html)). Although this approach incurs some additional computational cost, its overall impact on GPU memory and runtime is minimal.
>
> ---
>
> Q2. Why is a higher effective rank better? Both GeLU and SwiGLU have lower effective ranks than ReLU, but they achieve better performance.
>
> A2. A higher rank in the weight matrices generally indicates a greater capacity for representing complex patterns within the data. However, it is crucial to note that a higher effective rank is a necessary but not sufficient condition for neural networks to achieve better performance. Our experimental results serve as supplementary evidence, demonstrating that PolyCom enables neural networks to capture more intricate patterns.
>
> Regarding the specific case of ReLU as a counterexample, one plausible explanation is that ReLU discards part of the information in activations (values less than 0). This forces the backward gradient flow to encourage the weights to learn more complex patterns, ensuring that the activations after passing through ReLU retain sufficient information. Consequently, the weights in ReLU networks exhibit a larger effective rank.
>
> Thus, while effective rank provides insight into model expressivity, other factors, such as the characteristics of the activation function and its impact on gradient dynamics, also play critical roles in determining performance.
>
> ---
>
> Q3. It's not clear that PolyNorm MoE has lower layer-wise cosine similarity in Figure 7 compared to SwiGLU MoE.
>
> A3.  Comparing the rectangles in the **lower-left** (or upper-right) corner of Figure 7(c) and Figure 7(d), it is evident that the color corresponding to PolyNorm is noticeably redder than that of SwiGLU. This indicates that PolyNorm’s representations for layers 0–8 and 9–15 are less similar compared to SwiGLU’s.
>
> Additionally, examining the **lower-right** corner of Figure 7(c) and Figure 7(d), we observe that PolyNorm has fewer blue squares. This suggests that the representations across layers 9–15 in PolyNorm are less similar than those in SwiGLU. These observations collectively demonstrate that PolyNorm exhibits lower layer-wise cosine similarity, particularly in the deeper layers, thereby supporting our claims.
>
> ---

---

> > ### Comment · Reviewer_NayG · 2024-11-25
> >
> > I thank the authors for the clarifications. I recommend including the details on memory overhead and the use of gradient checkpointing in the paper, as these are important practical considerations. I still don't find the argument convincing regarding the necessity of a high effective rank. What is the evidence behind the claims that 1) "a higher rank in the weight matrices generally indicates a greater capacity for representing complex patterns within the data" and 2) "a higher effective rank is a necessary but not sufficient condition for neural networks to achieve better performance"? Furthermore, since GeLU and SwiGLU lead to lower effective ranks than ReLU, 2) would imply they cannot outperform ReLU, which is not the case.

---

> ### Author Response · Authors · 2024-11-20
> **Response to Reviewer NayG (2/2)**
>
> Q4.  Can you show how PolyCom affects the scaling laws of loss v.s. model size or training computed by, for example, training additional smaller models?
>
> A4.  The table below (also visualized in Figure 13, Appendix H) summarizes the training loss for dense models with SwiGLU, PolyReLU, and PolyNorm activations across a range of model sizes from 110M to 1.3B parameters. It is evident that both PolyReLU and PolyNorm consistently outperform SwiGLU across all model sizes.
>
> **Scaling Law Details:**
>
> |  | 110M | 226M | 502M | 1.3B |
> | --- | --- | --- | --- | --- |
> | SwiGLU | 2.719 | 2.535 | 2.379 | 2.187 |
> | PolyReLU | 2.710 | 2.522 | 2.360 | 2.173 |
> | PolyNorm | 2.700 | 2.514 | 2.353 | 2.168 |
>
> The model configurations for the scaling experiments are detailed below. All models employed the same hyperparameters as those specified for 1B dense models (see Table 7). Models with 110M, 226M, and 502M parameters were trained on a corpus of 200 billion tokens.
>
> | Params | Hidden size | Context Length | Intermediate size | Attention heads | Hidden Layers |
> | --- | --- | --- | --- | --- | --- |
> | 110M | 768 | 2048 | 2048/3072 | 16 | 12 |
> | 226M | 1024 | 2048 | 2560/3840 | 16 | 16 |
> | 502M | 1536 | 2048 | 4096/6144 | 16 | 16 |
> | 1.3B | 2048 | 4096 | 5504/8256 | 16 | 24 |
>
> This scaling law experiment demonstrates that PolyReLU and PolyNorm provide consistent improvements over SwiGLU as model sizes increase, confirming their benefits across different scales.

---

> ### Author Response · Authors · 2024-11-25
>
> ## Issue regarding the flops and memory overhead
> Thank you for the valuable suggestions. We have updated the previous table and will include it in the appendix later, which encompasses the version using gradient checkpointing (this is the configuration we actually used during training). It is important to note that the overhead and proportion often vary for different model sizes, so we have provided the corresponding formulas directly and take $H=1024, B=4, S=4096$, using BF16 precision as an example.
>
> - without gradient checkpointing:
>
> | Method | ReLU | GeLU | SwiGLU | ReLU$^2$ | 3rd-order PolyNorm | 3rd-order PolyReLU |
> | --- | --- | --- | --- | --- | --- | --- |
> | Intermediate Size | 4H | 4H | 8/3H | 4H | 4H | 4H |
> | FLOPs for activation | 4BSH |72BSH |112/3BSH |8BSH |72BSH |40BSH|
> | FLOPs ratio | 1/(6H)=0.016% | 3/H=0.29% | 14/(9H)=0.15% | 1/(3H)=0.032% | 3/H=0.29% | 5/(3H)=0.16% |
> | Memory Overhead | 4BSH=128MB | 10BSH=320MB | 8BSH=256MB | 8BSH=256MB | 12BSH=384MB |  8BSH=256MB |
>
> - with gradient checkpointing:
>
> | Method | ReLU | GeLU | SwiGLU | ReLU$^2$ | 3rd-order PolyNorm | 3rd-order PolyReLU |
> | --- | --- | --- | --- | --- | --- | --- |
> | Intermediate Size | 4H | 4H | 8/3H | 4H | 4H | 4H |
> | FLOPs for activation | 8BSH |144BSH |224/3BSH |16BSH |144BSH |80BSH|
> | FLOPs ratio (H=1024) | 1/(3H)=0.033% | 6/H=0.59% | 28/(9H)=0.30% | 2/(3H)=0.065% | 6/H=0.59% | 10/(3H)=0.33% |
> | Memory Overhead | 0 | 0| 0|  0 | 0 |  0 |
>
>
>
> The corresponding code (with gradient checkpointing) is as follows:
>
> - For PolyNorm
> ```python
> import torch
> from torch.utils.checkpoint import checkpoint
> import torch.nn.functional as F
>
> def _norm(x, eps=1e-6):
>     return x * torch.rsqrt(x.pow(2).mean(-1, keepdim=True) + eps)
>
> def _poly_norm(x, weight, bias, order=3):
>     return sum(weight[i] * _norm(x ** (i+1)) for i in range(order)) + bias
>
> class PolyNorm(torch.nn.Module):
>     def __init__(self):
>         super(PolyNorm, self).__init__()
>         self.weight = torch.nn.Parameter(torch.ones(3) / 3)
>         self.bias = torch.nn.Parameter(torch.zeros(1))
>
>     def forward(self, x, checkpointing=True):
>         if checkpointing:
>             return checkpoint(_poly_norm, x, self.weight, self.bias, use_reentrant=False)
>         return _poly_norm(x, self.weight, self.bias)
> ```
>
> - For PolyReLU
> ```python
> def _poly(x, weight, bias, order=3):
>     return sum(weight[i] * (x ** (i+1)) for i in range(order)) + bias
>
> class PolyReLU(torch.nn.Module):
>     def __init__(self):
>         super(PolyReLU, self).__init__()
>         self.weight = torch.nn.Parameter(torch.ones(3) / 3)
>         self.bias = torch.nn.Parameter(torch.zeros(1))
>
>     def forward(self, x, checkpointing=True):
>         x = F.relu(x)
>         if checkpointing:
>             return checkpoint(_poly, x, self.weight, self.bias, use_reentrant=False)
>         return _poly(x, self.weight, self.bias)
> ```

---

> > ### Author Response · Authors · 2024-11-25
> >
> > ## Issue regarding the effective rank
> > **Q1: Why do large effective ranks often lead to better performance?**
> >
> > A1: Mathematically, matrices with larger ranks typically encode more effective information, which translates to higher parameter efficiency. Furthermore, as analyzed in Sec. 4.1.1 of the paper "[Spectral Normalization for Generative Adversarial Networks](https://arxiv.org/abs/1802.05957)", when the singular values of model weights are more evenly distributed (a property aligned with a larger effective rank as defined in our paper), the feature space becomes broader and can capture more complex patterns, often resulting in better performance.
> >
> > Additionally, from our past experience in training large language models (LLMs), a larger effective rank generally correlates with improved downstream performance, assuming other factors remain constant. While effective rank serves as a valuable auxiliary metric alongside indicators like loss or perplexity, it is not the sole determinant of performance. A notable exception is ReLU, as discussed in the following Q2-A2 section.
> >
> >
> > **Q2: Why do GeLU and SwiGLU have lower effective ranks than ReLU?**
> >
> > A2: This is indeed an interesting question that deserves further analysis, which we will also attempt to explore in future research work.
> >
> > We posit that the distinctive rank of the ReLU activation is associated with the sparsity of ReLU-based models. This notion is further elaborated in the work "[Deja Vu: Contextual Sparsity for Efficient LLMs at Inference Time](https://arxiv.org/abs/2310.17157)", wherein a ReLU-based model can attain a contextual sparsity of 85%, implying that merely 15% of the activations are non-zero and distinct across varying inputs. Consequently, to maximize information preservation, the backpropagation optimization process tends to learn a larger effective rank for the weight parameters. In contrast, activation functions such as GeLU, SwiGLU, PolyNorm, and PolyReLU exhibit markedly lower non-zero components and sparse activations (approaching 0%), resulting in a different pattern.
> >
> > Due to the peculiar properties of ReLU-based models, we will elaborate further on this aspect within the paper and exclude the specific case of ReLU from the figures presented.

---

### Official Review · Reviewer_aNH6 · 2024-11-04

**Soundness:** 2
**Presentation:** 3
**Contribution:** 2
**Rating:** 6
**Confidence:** 3

**Summary:**

This work presents a method to extend activation functions using polynomials.
Two examples of these extended activation functions are introduced: PolyReLU and PolyNorm.
PolyReLU is shown to achieve an optimal approximation rate in Sobolev spaces.
Language modelling experiments indicate that the proposed activation functions accelerate training.

**Strengths:**

- (clarity) Overall, the paper is clearly written and easy to follow.
 - (originality) The proposed activation functions are novel (to the best of my knowledge).

**Weaknesses:**

- (quality) The proof of lemma 2 (and therefore also lemma 2) seems to be incorrect.
   It seems like the rank (and number of polynomials) from lemma 3.4 in (Telgarsky, 2017) have been ignored.
   Furthermore, it is unclear why the minimum has disappeared because even $\ln(1/x)^2 < \ln(1/x)$ does **not** hold for $x \in (0, 1)$.
   Also, there is no argument for why the addition of the ReLU function in the proof does not change the size of the overall network.
 - (quality/significance) The lower bound in theorem 2 does not make sense.
   If the PolyReLU network happens to have PolyReLU parameters $a_1 = 1$ and $\forall i \neq 1 : a_i = 0$,
   it should be possible to model it exactly with a ReLU network of the same size.
   Similarly, I do not see how the upper bound in theorem 2 would make sense.
   It should always be possible to make the network arbitrarily large by adding neurons for which all incoming and outgoing weights are zero.
 - (clarity/quality) The polynomial activation functions seem to introduce additional learnable parameters.
   However, the paper never explicitly states this anywhere.
   Futhermore, there is no discussion on whether/how the number of parameters of the different models was controlled.
   This possibly introduces a capacity advantage for the models with PolyCom functions compared to the baselines,
   leading to an unfair comparison.
 - (clarity/quality) There is no discussion on how the hyper-parameters were found.
   Furthermore, it seems like the same hyper-parameters were used for every model.
   If these hyper-parameters were tuned on the proposed models, this would be an unfair comparison.
 - (quality) The authors claim that the activation functions are able to capture higher-order interactions.
   However, based on the training curves (e.g. figure 3), none of the models were trained to convergence.
   As a result, it is not possible to conclude anything concerning the complexity of the interactions that can be learned.
   After all, it might be that the same interactions are captured for both models when converged, but one captures them "faster".
   As a result, these experiments do not confirm the stated claims.
 - (quality/significance) There is no discussion on the runtime overhead of the introduced functions.
   If the PolyCom functions introduce too much overhead, they might not be practically useful.
   Also, it would be useful to include a comparison performance for a given compute budget.
 - (clarity) It is not clear why the proposed activation functions should be especially suited for language modelling.
   I suspect that this should also work for models that do not require that much compute and allow for more extensive experiments.

###### Minor Comments
 - There seem to be some type-setting issues with equation (7) (Thereom 1).
 - Could it be that there is a superscript $d$ missing for the space of $\boldsymbol{m}$ on line 869-870?

**Questions:**

1. Does theorem 2 make sense?
2. What is happening in the proof of lemma 2?
3. Do the PolyCom functions introduce additional learnable parameters?
4. Do the different models in the comparison have the same number of parameters?
5. How were the hyperparameters tuned?
6. How do the models compare when trained to convergence?
7. What does the performance look like as a function of training budget?
8. Does this method work also provide benefits when used outside of language modelling?

---

> ### Author Response · Authors · 2024-11-20
> **Response to Reviewer aNH6 (1/3)**
>
> Thanks for your time and constructive comments. We address your concerns point by point below:
>
> ---
>
> Q1. Does theorem 2 make sense?
>
> A1.   We are afraid that you may have misunderstood the meaning of Theorem 2. We restate Theorem 2 for better clarity.
>
> **Theorem 2.** For a PolyReLU network $g$ of depth $L$, width $K$, and PolyReLU activation of order and Lipschitz constant $\alpha$. Suppose each neuron computes $x \mapsto PolyReLU(a^\top x +b)$  with the parameters pair $(a,b)$ satisfies $\|a\|_1+b\leq 1, PolyReLU:[-1,1]\rightarrow [-1,1]$ (a, b, and PolyReLU are possibly distinct across neurons). For any given $\epsilon \in (0,1)$, there exists a ReLU network $f:[-1,1]^d \rightarrow [-1,1]$ of size
>
> $$O\left(LK\ln^2\left(\frac{L\alpha^L}{\epsilon}\right)\right),$$
>
> such that  $\max_{x \in [-1,1]^d} |f(x)-g(x)| < \epsilon$.
>
> Conversely, there exist some PolyReLU networks that cannot be approximated within error tolerance $\epsilon$ by any ReLU network with a size less than  $\Omega \left(KL\ln\left(\frac{1}{\epsilon}\right)\right)$.
>
> **Clarifications:**
>
> 1. The lower bound asserts that **there exist some** PolyReLU networks that can not be approximated by any ReLU network with a size less than  $\Omega \left(KL\ln\left(\frac{1}{\epsilon}\right)\right)$. This does not imply all PolyReLU networks exhibit this behavior, hence the example provided does not contradict our claim.
> 2. For the upper bound, we mean that for **any** PolyReLU networks $g$
> , there is a ReLU network $f$
>  of size $O\left(LK\ln^2\left(\frac{L\alpha^L}{\epsilon}\right)\right)$ that can approximate f within error tolerance $\epsilon$. The upper bound provides a constructive guarantee that a ReLU network of the specified size can approximate any PolyReLU network to the desired accuracy. Enlarging the network size unnecessarily weakens the bound’s tightness and is therefore irrelevant to the result.
>
> ---
>
> Q2.  Clarifications regarding the proof of Lemma 2.
>
> A2.  For the proof of Lemma 2, we are afraid that you may have misunderstood.  We respond to your concerns point by point.
>
> 1. The size of PolyReLU network, denoted as $g$, can indeed be written as $O(\ln^2(\frac{r}{\epsilon}))$. However, since $r$ is a fixed positive constant, we focus on the approximation error $\epsilon$ and incorporate $r$ into the $O$-notation. Thus, $O(\ln^2(\frac{r}{\epsilon}))$ simplifies to $O(\ln^2(\frac{1}{\epsilon}))$.
> 2. From Lemma 3.4 in (Telgarsky, 2017), the size of the ReLU network $f$ satisfying the conditions is
>
>     $$O(\min(r\ln(r/\epsilon), \ln^2(r/\epsilon))).$$
>
>     Using the fact  $\min\{a,b\}\leq b$ (based on this, the minimum disappears), we get
>
>     $$O(\min(r\ln(r/\epsilon), \ln^2(r/\epsilon))) \leq O( \ln^2(r/\epsilon)).$$
>
>     Incorporating $r$ into the $O$-notation, we immediately get the size of  the ReLU network $f$ is $O(\ln^2(1/\epsilon))$.
>
> 3. Adding the ReLU activation does not introduce additional parameters or modify the layer sizes. Thus, the overall network size remains unchanged.
>
> ---
>
> Q3. Do the PolyCom functions introduce additional learnable parameters?
>
> A3.  Yes, PolyCom introduces additional learnable parameters, as explicitly stated in Section 2 (lines 134–136). For instance, a third-order PolyCom incorporates an additional 4L parameters in a transformer with L layers. For a 1B dense model with 24 layers, the supplementary parameter count amounts to 96 (in contrast to the total of 1.3 billion parameters). Similarly, for the MoE model with 16 layers, the additional count is 48 (compared to 6.9 billion total parameters). These increments are quite small and can be considered negligible.

---

> ### Author Response · Authors · 2024-11-20
> **Response to Reviewer aNH6 (2/3)**
>
> ---
>
> Q4. Do the different models in the comparison have the same number of parameters?
>
> A4. **All compared models were designed to maintain an equivalent parameter count. (1.3 billion for the dense model and 6.9 billion for the MoE mode)**. All compared models were designed to maintain an equivalent parameter count for fair comparison. For instance, SwiGLU’s intermediate size is adjusted to two-thirds that of other activations. More details are provided in Tables 5 and 6 (Appendix D).
>
> ---
>
> Q5. How were the hyperparameters tuned?
>
> A5. Hyperparameters were not tuned specifically for our experiments. We adopted hyperparameters similar to those used in LLaMA 2 [1] for the dense model (note that not all hyperparameters were explicitly reported in their work) and employed the default hyperparameters from OLMoE [2] for the MoE models. The following table summarizes the hyperparameters:
>
> |  | 1B dense model | MoE-1B-7B |
> | --- | --- | --- |
> | Optimizer | AdamW | AdamW |
> | Learning rate (LR) | $3\times10^{-4}$ | $4\times10^{-4}$ |
> | minimum LR | $3\times10^{-5}$ | $5\times10^{-5}$ |
> | LR schedule | cosine | cosine |
> | Weight decay | 0.1 | 0.1 |
> | $\beta_1$ | 0.9 | 0.9 |
> | $\beta_2$ | 0.95 | 0.95 |
> | Gradient clipping | 1 | 1 |
> | Warmup tokens | 620000000 | - |
> | Warmup steps | - | 2000 |
> | Init distribution | normal | trunc normal |
> | Init std | $1/(2d)$ | $1/(2d$) |
> | Init trunc | - | 3 $\times$ std |
> | Load balancing loss weight | - | 0.01 |
> | Router z-loss weight | - | 0.001 |
>
> References:
>
> [1] Hugo Touvron, et al. Llama 2: Open foundation and fine-tuned chat models.
>
> [2] Niklas Muennighoff,  et al.  Olmoe: Open mixture-of-experts language models, 2024
>
> ---
>
> Q5.  How do the models compare when trained to convergence?
>
> A5. Achieving complete convergence in pretraining large language models poses a significant challenge due to the immense computational resources required. For example, Meta’s training of the LLaMA-3.2 1B model (https://huggingface.co/meta-llama/Llama-3.2-1B) involved processing 9T tokens and consuming 370,000 H100-GPU hours, which translates to **64 H100 GPUs for 8 months** or **64 A100 GPUs for 2 years**. For most research aiming at pretraining improvements, the performance gap between models typically **stabilizes** after reaching a certain threshold of training corpus size. To illustrate this, we conducted additional experiments on dense models by extending the training corpus from 250 billion to 400 billion tokens. These experiments utilized 32 A100 GPUs over more than 10 days. The following tables present the training loss and validation perplexity (PPL) of various models. More detailed analyses can be found in Figure 11 (Appendix F).
>
> Our results demonstrate that models using **PolyReLU** and **PolyNorm** consistently outperform those with **SwiGLU**, even with larger datasets. For large language model pretraining, a training loss difference greater than 0.015 is considered substantial. Moreover, the improvements from **PolyReLU** and **PolyNorm** have remained **stable** beyond the 200 billion token mark. These results indicate that **PolyReLU** and **PolyNorm** not only improve training dynamics but also consistently yield better validation performance compared to **SwiGLU**. This trend suggests that their advantages persist across different scales of training corpus.
>
> | Training Loss | 100 billion | 200 billion | 300 billion | 400 billion |
> | --- | --- | --- | --- | --- |
> | SwiGLU | 2.258 | 2.202 | 2.174 | 2.158 |
> | PolyReLU | 2.242 | 2.190 | 2.163 | 2.148(-0.01) |
> | PolyNorm | **2.233** | **2.182** | **2.158** | **2.143(-0.015)** |
>
> | Validation PPL | 100 billion | 200 billion | 300 billion | 400 billion |
> | --- | --- | --- | --- | --- |
> | SwiGLU | 3.354 | 3.225 | 3.170 | 3.111 |
> | PolyReLU | 3.309 | 3.193 | 3.141 | 3.086(-0.025) |
> | PolyNorm | **3.298** | **3.183** | **3.133** | **3.074（-0.037）** |
>
> ---

---

> ### Author Response · Authors · 2024-11-20
> **Response to Reviewer aNH6 (3/3)**
>
> Q6. There is no discussion on the runtime overhead of the introduced functions. What does the performance look like as a function of training budget?
>
> A6. We analyze the runtime overhead introduced by the activation functions using a typical feedforward network (FFN) with input tensor $x\in \mathbb{R}^{B \times S \times H}$, where $B$，$S$ and $H$ represent the batch size, sequence length, and hidden size, respectively. The relationship between computational FLOPs and model parameters can generally be regarded as proportional (as discussed in [Eleuther AI’s transformer math](https://blog.eleuther.ai/transformer-math/)). Below, we estimate the proportion of the computational cost incurred by activation function calculations within the total computational cost of the FFN matrix computations ($24BSH^2$). The FLOPs ratio is calculated as:
>
> $$\text{FLOPs ratio} = \frac{\text{FLOPs for activation}}{24BSH^2}$$
>
> The results are summarized in the following table:
>
> | Method | ReLU | GeLU | SwiGLU | ReLU^2 | 3rd-order PolyNorm | 3rd-order PolyReLU |
> | --- | --- | --- | --- | --- | --- | --- |
> | Intermediate Size | 4H | 4H | 8/3H | 4H | 4H | 4H |
> | FLOPs for activation | 4BSH |72BSH |112/3BSH |8BSH |72BSH |40BSH|
> | FLOPs ratio (H=1024) | 1/(6H)=0.016% | 3/H=0.29% | 14/(9H)=0.15% | 1/(3H)=0.032% | 3/H=0.29% | 5/(3H)=0.16% |
>
> Note:
> - We assume that the scale of the input tensor is set to [-1, 1]. In this case, the FLOPs for both tanh and exp are approximately 10 each.
> - For a fair comparison, the intermediate size for SwiGLU is set to $8/3H$ to ensure a consistent total parameter count across all methods.
> - To reduce memory overhead during large language model pretraining, we typically employ gradient checkpointing (refer to [PyTorch Docs](https://pytorch.org/docs/stable/checkpoint.html)). Although this approach incurs some additional computational cost, its overall impact on GPU memory and runtime is minimal.
>
> Hence, the overhead is acceptable and there is not much difference in the amount of training budget between them.
>
> ---
>
> Q7.  Does this method work also provide benefits when used outside of language modelling?
>
> A7.  To evaluate the effectiveness of PolyCom beyond language modeling, we trained ResNet50 on ImageNet following the settings of timm [1]. For comparison, we replaced the ReLU activation in ResNet50 with PolyCom and reported the training loss and top-1/top-5 accuracy on the evaluation set, as shown in the table below. The results demonstrate that PolyCom outperforms ReLU by a significant margin in terms of training loss, top-1 accuracy, and top-5 accuracy.
>
> | Training Loss ↓ | 50 epoch | 100 epoch | 150 epoch | 200 epoch |
> | --- | --- | --- | --- | --- |
> | ResNet50(ReLU) | 2.586 | 2.342 | 2.203 | 2.121 |
> | ResNet50(PolyCom) | **2.531**(-0.055) | **2.259**(-0.083) | **2.117**(-0.086) | **2.026**(-0.095) |
>
> | Evaluation acc@1/acc@5 ↑ | 50 epoch | 100 epoch | 150 epoch | 200 epoch |
> | --- | --- | --- | --- | --- |
> | ResNet50(ReLU) | 70.089/89.510 | 72.971/91.108 | 74.197/91.736 | 74.913/92.031 |
> | ResNet50(PolyCom) | **71.502/90.294**(+1.413/+0.784) | **73.530/91.581**(+0.559/+0.473) | **74.685/91.978**(+0.488/+0.242) | **75.117/92.099**(+0.204/+0.068) |
>
> References:
>
> [1]. Ross Wightman. PyTorch Image Models. https://github.com/rwightman/pytorch-image-models
>
> ---
>
> Thank you for highlighting the typos. We have corrected these errors and hope our explanations address your concerns. Please feel free to let us know if further clarification is needed.

---

> > ### Author Response · Authors · 2024-11-23
> > **Follow-up on Review Feedback**
> >
> > Dear Reviewer aNH6:
> >
> > We sincerely appreciate the time and effort you have devoted to providing thoughtful and constructive feedback on our submission. Your insights have been incredibly valuable in helping us refine our work.
> >
> > As the public discussion phase is nearing its conclusion on November 26th, we wanted to kindly follow up to see if there are any additional questions, concerns, or points that we could clarify or address to further assist with your review process. We are more than happy to provide any additional information or details you might need.

---

> > > ### Comment · Reviewer_aNH6 · 2024-11-25
> > > **Sorry for the last-minute reply.**
> > >
> > > Thank you for addressing my concerns.
> > >
> > > A1: I don't know why I got so confused about the upper bound in theorem 2.
> > > The reformulation concerning the lower bound makes more sense to me than the original version.
> > >
> > > A2:
> > >  1. According to this logic also $K$, $L$ (and $\alpha$) could be incorporated in the big-O notation. Why is $r$ handled differently?
> > >  2. In that case, why choose $\ln(1/\epsilon)^2$ instead of $\ln(1/\epsilon)$?
> > >  3. I realise that it is quite trivial, but I think it is important for the proof.
> > >
> > > A3: I thought I saw it somewhere, but couldn't find it when writing up the review. It would be good to make this more prominent in the paper. Also, I understand where the $4L$ figure comes from in case of PolyNorm, but doesn't the PolyReLU introduce $4LK$ parameters? After all, in contrast to PolyNorm, PolyReLU does not seem to be a vector function.
> > >
> > > A4: OK
> > >
> > > A5 (1): OK
> > >
> > > A5 (2): Still, these experiments only prove that PolyCom functions are easier to train.
> > > It would have been interesting to know if final models end up being better or whether the difference disappears eventually. The additional plots seem to suggest that the performance difference diminishes as training proceeds. I think it would have been more interesting to take a much smaller model and provide an example on a small scale experiment.
> > >
> > > A6: OK, does this mean that it is safe to assume that the performance benefits that can be seen in the loss curves does not disappear when plotted in terms of runtime (instead of number of tokens)?
> > >
> > > A7: Thank you. Do you plan on including these results in the paper? Have these models been trained to convergence? Do these learning curves in this setting look similar to those presented in the paper?
> > >
> > > I apologise once more for the last-minute reply, but I hope you still have time to help me understand this contribution. I will take into account your response(s) as well as the other reviews and plan to update my score accordingly.

---

> > > > ### Author Response · Authors · 2024-11-26
> > > > **Futher Reply**
> > > >
> > > > Q1.  I don't know why I got so confused about the upper bound in theorem 2.
> > > >
> > > > A1. The upper bound in Theorem 2 indicates that for any PolyReLU network $g$, there exists a ReLU network $f$ of size $O\left(LK\ln^2\left(\frac{L\alpha^L}{\epsilon}\right)\right)$ (or alternately $O(\min(LKr\ln(r/\epsilon), LKr\ln^2(Lr\alpha^L/\epsilon))$, as explained in below A2) that can approximate $g$ within an error $\epsilon$. For the upper bound, we want it as tight (i.e. small) as possible. it is pointless to enlarge the ReLU network arbitrarily by adding neurons with zero incoming and outgoing weights (as you suggested) to approximate the PolyReLU network $g$.
> > > >
> > > > ---
> > > > Q2. More about Lemma 2 and Theorem 2.
> > > >
> > > > A2.   This concern primarily relates to **a matter of presentation rather than correctness**. Without incorporating $r$ into the big-O notation and applying further inequality relaxations, the upper bound in Lemma 2 can indeed be expressed as $O(\min(r\ln(r/\epsilon), \ln^2(r/\epsilon)))$, and the upper bound in Theorem 2 as $O(\min(LKr\ln(r/\epsilon), LKr\ln^2(Lr\alpha^L/\epsilon)))$. Both Lemma 2 and Theorem 2 remain correct under these representations. This form of presenting the upper bound might be more acceptable to you. Our original approach aimed to present the results more straightforwardly.
> > > >
> > > > ---
> > > > Q3.  Does PolyReLU introduce 4LK parameters?
> > > >
> > > > A3.  No, the third-order PolyReLU introduces only 4L additional parameters in a transformer with  L  layers. Each layer contains a single PolyReLU, and each third-order PolyReLU contributes just 4 parameters. For further details, please refer to the code provided in Global Response to the Computational Overhead and Memory Footprint.
> > > >
> > > > ---
> > > > Q4.  Still, these experiments only prove that PolyCom functions are easier to train. It would have been interesting to know if the final models end up being better or whether the difference disappears eventually.
> > > >
> > > > A4. The core focus of our paper is to investigate **whether there exists a more efficient activation function** that can achieve **superior performance under a relatively reasonable budget** when training **large language models**.
> > > >
> > > > Our answer is: **Yes**.
> > > >
> > > > We conducted experiments on dense models ranging from 100M to 1B parameters, as well as MoE models with 7B total parameters, which is uncommon in LLM pretraining research [1][2]. Significant performance improvements were observed across these experiments. From these results, we are highly confident that our activation function exhibits stronger fitting capabilities and can scale to larger model sizes.
> > > > It's perhaps an interesting question whether training LLMs with infinite training budgets and infinite corpora could diminish the performance gains brought about by model architecture adjustments. However, this scenario is largely impractical in real-world LLM settings.
> > > >
> > > > Regarding the experiments with smaller models, as mentioned in [other comments](https://openreview.net/forum?id=CbpWPbYHuv&noteId=2m3A1uQqZj), we have indeed conducted experiments on 100M models, and the results are consistent with our existing conclusions. Again, the convergence behavior under an infinite budget is NOT our primary concern.
> > > >
> > > > Furthermore, it is essential to emphasize that when improving model architectures, the focus should be on larger models and scalability. Because many improvements that are effective for smaller models may fail to translate to larger ones, and the model size inherently limits its intelligence.
> > > >
> > > > Additionally, "ease of training" is one of the rarest and most valuable characteristics in LLM training. Imagine a structure that can achieve a 1.5x acceleration in convergence while keeping the same performance – this would be highly meaningful.
> > > >
> > > > [1] [Switch Transformers: Scaling to Trillion Parameter Models with Simple and Efficient Sparsity](https://arxiv.org/pdf/2101.03961). 180B training tokens
> > > >
> > > > [2] [OPT: Open Pre-trained Transformer Language Models](https://arxiv.org/pdf/2205.01068). 180B training tokens
> > > >
> > > > ---
> > > > Q5. Does this mean that it is safe to assume that the performance benefits that can be seen in the loss curves do not disappear when plotted in terms of runtime?
> > > >
> > > > A5. Yes, the computation of the activation function in LLMs accounts for a small fraction of the overall computation, as discussed in Global Response to the Computational Overhead and Memory Footprint. These minimal runtime differences do not diminish the observed training benefits.
> > > >
> > > >
> > > > ---
> > > > Q6. Do you plan on including these results in the paper? Have these models been trained to convergence? Do these learning curves in this setting look similar to those presented in the paper?
> > > >
> > > > A6. We will include these results in the later version. For training ResNet-50 on ImageNet-1k, 200 epochs are sufficient for convergence. From the table in previous A7, we observe significant differences in both training loss and test accuracy across the compared methods, which align with the trends presented in the paper.

---

> > > > ### Author Response · Authors · 2024-11-29
> > > >
> > > > Dear Reviewer aNH6:
> > > >
> > > > Thank you for your valuable feedback on our submission. We have carefully considered the inquiries and provided detailed responses to the questions. We are kindly following up to check if there are any further clarifications needed.
> > > >
> > > > We appreciate your time and effort in reviewing our paper and remain available for any additional questions.

---

> > > > ### Author Response · Authors · 2024-11-30
> > > >
> > > > Dear Reviewer aNH6,
> > > >
> > > > As the review deadline (**December 2nd**) approaches, we sincerely appreciate the time and effort you’ve dedicated to evaluating our responses. We kindly follow up to confirm whether our replies have adequately addressed your questions or if there are any additional clarifications we can provide.
> > > >
> > > > We would also be truly grateful if you could consider re-evaluating the paper’s rating based on the updated responses.

---

> ### Author Response · Authors · 2024-12-02
>
> Dear Reviewer aNH6,
>
> As the review deadline (**December 2nd**) approaches, we sincerely thank you for your time and effort in reviewing our work. We kindly follow up to confirm if our responses have addressed your concerns and whether any further clarification is needed.
>
> We would greatly appreciate it if you could re-evaluate the paper’s rating based on the updated responses.

---

> > ### Comment · Reviewer_aNH6 · 2024-12-02
> >
> > In accordance with the other reviewers, I decided to increase my score towards acceptance.
> > Most of my concerns (and misunderstandings) have been addressed and the empirical performance improvements look promising.
> > However, the disconnect between theory and practice remains, because models are not trained until convergence and therefore, the theoretically claimed increase in expressivity of the resulting models is not properly tested.
> >
> > PS: I would urge the authors to trust the reviewers/area chairs/review process more and refrain from sending so much spam for future submissions.

---

> > > ### Author Response · Authors · 2024-12-02
> > >
> > > Thank you for your thoughtful feedback and for increasing your score. We are glad that most of your concerns have been addressed.
> > >
> > > We apologize if our communication during the review process was excessive and hope you can understand our intention to clarify any misunderstandings. We greatly value the efforts of reviewers and area chairs and will improve in future submissions.
> > >
> > > Thank you again for your constructive feedback and support.

---

### Author Response · Authors · 2024-11-20
**A Summary of Paper Updates**

We sincerely thank all reviewers for their constructive feedback. Based on your valuable suggestions, we have refined the paper to enhance clarity and rigor. The key updates include:

- Section 3.2: Restated the lower bound portion of Theorem 2 for improved clarity.
- Appendix B: Expanded discussion on the optimal approximation rate.
- Appendix D2: Added hyperparameters for the 1B dense model and MoE-1B-7B in Table 7.
- Appendix E: Introduced computational complexity analysis.
- Appendix H: Presented scaling curves for different models.

We hope these adjustments meet your expectations and contribute to the improved readability and comprehension of our paper. If there are any further questions or suggestions, please feel free to let us know. We are looking forward to providing additional clarification and discussion.

---

> ### Author Response · Authors · 2024-11-23
> **Follow-Up on Review Feedback**
>
> Dear Reviewers,
>
> Thank you once again for your valuable and constructive feedback on our submission. We deeply appreciate the time and effort you have dedicated to reviewing our work.
>
> We want to kindly ask if there are any additional questions or aspects of our submission that you would like us to clarify or elaborate on. We are more than happy to provide any further explanations to support your review process.

---

### Author Response · Authors · 2024-11-25
**Global Response to the Computational Overhead and Memory Footprint**

We would like to thank again to all the reviewers for their insightful comments!

One common question raised by the reviewers is: what are **the computational overhead and memory footprint** incurred by those activation functions?


To address this, we conducted a thorough analysis using a typical feedforward network with input tensor $x\in \mathbb{R}^{B \times S \times H}$, where $B$, $S$, and $H$ represent the batch size, sequence length, and hidden size, respectively. The relationship between computational FLOPs and transformer model parameters can generally be regarded as proportional (as discussed in [Eleuther AI’s transformer math](https://blog.eleuther.ai/transformer-math/)). Below, we estimate the proportion of the computational cost incurred by activation functions within the total computational cost of the FFN matrix computations ($24BSH^2$) (actually, when considering all transformer modules, the ratio would be even smaller).

The FLOPs ratio is:

$$\text{FLOPs ratio} = \frac{\text{FLOPs for activation}}{24BSH^2}$$

It is important to note that the overhead and proportion often vary for different model sizes, so we provide the corresponding formulas directly and take $H=1024, B=4 (\texttt{each\ device}), S=4096$, using BF16 precision as an example.

- without gradient checkpointing:

| Method               |     ReLU      |    GeLU     |    SwiGLU     |   ReLU$^2$    | 3rd-order PolyNorm | 3rd-order PolyReLU |
| -------------------- | :-----------: | :---------: | :-----------: | :-----------: | :----------------: | :----------------: |
| Intermediate Size    |      4H       |     4H      |     8/3H      |      4H       |         4H         |         4H         |
| FLOPs for activation |     4BSH      |    72BSH    |   112/3BSH    |     8BSH      |       72BSH        |       40BSH        |
| FLOPs ratio          | 1/(6H)=0.016% |  3/H=0.29%  | 14/(9H)=0.15% | 1/(3H)=0.032% |     3/H=0.29%      |    5/(3H)=0.16%    |
| Memory Overhead      |  4BSH=128MB   | 10BSH=320MB |  8BSH=256MB   |  8BSH=256MB   |    12BSH=384MB     |     8BSH=256MB     |

- with gradient checkpointing:

| Method               |     ReLU      |   GeLU    |    SwiGLU     |   ReLU$^2$    | 3rd-order PolyNorm | 3rd-order PolyReLU |
| -------------------- | :-----------: | :-------: | :-----------: | :-----------: | :----------------: | :----------------: |
| Intermediate Size    |      4H       |    4H     |     8/3H      |      4H       |         4H         |         4H         |
| FLOPs for activation |     8BSH      |  144BSH   |   224/3BSH    |     16BSH     |       144BSH       |       80BSH        |
| FLOPs ratio (H=1024) | 1/(3H)=0.033% | 6/H=0.59% | 28/(9H)=0.30% | 2/(3H)=0.065% |     6/H=0.59%      |   10/(3H)=0.33%    |
| Memory Overhead      |       0       |     0     |       0       |       0       |         0          |         0          |

Note:
- We assume the scale of the input is set to [-1, 1]. In this case, the FLOPs for both tanh and exp are approximately 10 each.
- For a fair comparison, the intermediate size for SwiGLU is set to $8/3H$ to ensure a consistent total parameter count.
- To reduce memory overhead during pretraining, we typically employ gradient checkpointing (refer to [PyTorch Docs](https://pytorch.org/docs/stable/checkpoint.html)). Although this approach incurs some additional computational cost, its overall impact on GPU memory and runtime is minimal.

Hence, **the overhead and memory footprint are acceptable and there is not much difference in the amount of training budget between them**.

Additionally, we have provided the code with gradient checkpointing as follows:

- For PolyNorm
```python
import torch
from torch.utils.checkpoint import checkpoint
import torch.nn.functional as F

def _norm(x, eps=1e-6):
    return x * torch.rsqrt(x.pow(2).mean(-1, keepdim=True) + eps)

def _poly_norm(x, weight, bias, order=3):
    return sum(weight[i] * _norm(x ** (i+1)) for i in range(order)) + bias

class PolyNorm(torch.nn.Module):
    def __init__(self):
        super(PolyNorm, self).__init__()
        self.weight = torch.nn.Parameter(torch.ones(3) / 3)
        self.bias = torch.nn.Parameter(torch.zeros(1))

    def forward(self, x, checkpointing=True):
        if checkpointing:
            return checkpoint(_poly_norm, x, self.weight, self.bias, use_reentrant=False)
        return _poly_norm(x, self.weight, self.bias)
```

- For PolyReLU
```python
def _poly(x, weight, bias, order=3):
    return sum(weight[i] * (x ** (i+1)) for i in range(order)) + bias

class PolyReLU(torch.nn.Module):
    def __init__(self):
        super(PolyReLU, self).__init__()
        self.weight = torch.nn.Parameter(torch.ones(3) / 3)
        self.bias = torch.nn.Parameter(torch.zeros(1))

    def forward(self, x, checkpointing=True):
        x = F.relu(x)
        if checkpointing:
            return checkpoint(_poly, x, self.weight, self.bias, use_reentrant=False)
        return _poly(x, self.weight, self.bias)
```

---

### Author Response · Authors · 2024-12-04

Dear Reviewers and Area Chairs,

We are truly grateful for the thorough insights and constructive feedback offered by the reviewers during the review process, as well as the valuable guidance from the Area Chairs. As we reach the end of the rebuttal phase, we would like to highlight the key contributions of our manuscript, "Polynomial Composition Activations: Unleashing the Dynamics of Large Language Models," and share our reflections on the discussions surrounding the feedback we received.

**Main Contributions**

We propose a novel activation function, PolyCom, which is a composition of polynomial and other types of functions. Furthermore, we introduce two instances of PolyCom: PolyReLU and PolyNorm, and detail their seamless integration into the transformer architecture. Our activation functions highlight the following key advantages:
1. **Comparable Computational Overhead and Memory Footprint**: the overhead and memory footprint are acceptable, and there is negligible difference in the training budget required compared to the widely used SwiGLU.
2. **Optimal Theoretical Approximation Rates**: Theoretically, we derive bounds on the number of trainable parameters required for PolyReLU networks to approximate ReLU networks, and vice versa. Moreover, we demonstrate that a PolyReLU network of size $O(\epsilon^{-d/n})$ can approximate any function in Sobolev spaces with error tolerance $\epsilon$, thereby achieving the optimal approximation rates.
3. **Better Convergence Speed**: Empirically, we validate the efficacy of this novel activation function on LLMs with 1B dense models and MoE models with 1B active and 7B total parameters. The results of both models reveal that PolyCom can attain a remarkable 1.5x convergence speedup when contrasted with SwiGLU.

These contributions were positively acknowledged by the reviewers:
- "...exhibit non-trivial performance gains for training language models with >1B parameters, even compared to strong baselines such as SwiGLU and squared ReLU." by Reviewer NayG.
- "...along with improved convergence rates in the learning curves with fixed parameter size model..." by Reviewer "TiU9"
- "...the empirical performance improvements appear promising..." by Reviewer aNH6
- "...has strong theoretical guarantees, ...and has an optimal approximation rate for general smooth functions in Sobolev spaces." by Reviewer NayG
- "...without adding more trainable parameters to the model..." by Reviewer NvaU


**Summary of Revisions**

In response to the constructive feedback from the reviewers, we have made several significant adjustments and additions:

**Addressing common responses:**
1. **Expanded Theoretical Clarifications:** Based on reviewers' comments, we clarified the theoretical contributions, particularly regarding Theorem 2 and Lemma 2, ensuring the distinction between supremum and upper bounds was accurately presented.
2. **Added Computational Complexity Analysis:** We provided detailed analyses of the runtime and memory overhead for the proposed activation functions, including FLOPs ratios and memory consumption, which were included in the revised appendix.
3. **Clarification on Training Stability:** We expanded the discussion on training stability for PolyReLU and PolyNorm, attributing stability to normalization operators and demonstrating consistent results across transformer-based and ResNet50 architectures. Furthermore, we incorporate some practical usage recommendations.
4. **Included Additional Experiments:** In response to requests for broader evaluations, we conducted experiments on ResNet50 for non-transformer settings and extended scaling law evaluations to compare performance across model sizes.

**Addressing individual responses:**
1. **Reviewer aNH6:** We addressed the misunderstandings about Theorem 2, explained impracticality of full convergence in LLM pretraining, and clarified minimal parameter increments for PolyReLU and PolyNorm.
2. **Reviewer NayG:** We provided FLOPs and memory analyses, discussed effective rank's role in expressivity, and added scaling law experiments showing consistent performance improvements.
3. **Reviewer NvaU:** We compared computational overhead and generalizability beyond transformers with new ResNet50 experiments; clarified Theorem 4.2 for optimal approximation rates.
4. **Reviewer TiU9:** We explained how PolyNorm mitigates instability in FP16/BF16 training and confirmed no exploding gradient issues due to normalization and clipping strategies.

We believe that we have addressed the reviewers' key concerns effectively through thoughtful revisions and clarifications. These improvements have strengthened the clarity, rigor, and overall quality of our paper.

Once more, we extend our heartfelt thanks to the reviewers and area chairs for their constructive feedback and valuable guidance. Your time and effort have been pivotal in refining our work, and we are truly grateful for your support.

Best regards,

Authors of Submission #10038

---

### Meta-Review · Area_Chair_WwqA · 2024-12-21

**Metareview:**

**Summary**

This work introduces a novel activation function called PolyCom, which is a composition of polynomials and other types of functions featuring specific instances like PolyReLU and PolyNorm. These are seamlessly integrated into transformer architectures, offering several key benefits. They maintain a comparable computational overhead and memory footprint to the widely used SwiGLU activation function while providing optimal theoretical approximation rates. PolyReLU, for example, is capable of achieving the optimal approximation rate in Sobolev spaces. Empirical tests on large language models (LLMs) and Mixture of Experts (MoE) models demonstrate that PolyCom can significantly enhance training efficiency, achieving up to 1.5 times faster convergence compared to SwiGLU.

**Strengths**


* The paper is well-structured and well-written, and the results are presented concisely.
* The core idea of the paper on devising activation functions with better expressivity without adding more trainable parameters to the model is significant, useful, and interesting.
* The construction/definition of PolyCom is very flexible and serves as a good base for future experimentation with other activation variants beyond the specific instantiations of PolyReLU and PolyNorm.
* Extensive experiments on downstream tasks demonstrate improved convergence rates in learning curves with fixed model parameter sizes. These results underscore the potential of the proposed activation function and substantiate the theoretical claims made in the paper.

**Weaknesses**

* The original submission lacked computational complexity analysis for the proposed activations.
* Without proper normalization layers, the proposed activations could lead to exploding gradients. However, this won't be an issue in transformer networks that utilize layer normalization.

**Conclusion**

The paper received positive feedback from all reviewers. The weaknesses highlighted by the reviewers were effectively addressed by the authors in their revisions, providing a more comprehensive understanding of the proposed work. After thoroughly reviewing the paper, the feedback from the reviewers, and the authors' rebuttal, I agree with the majority opinion and vote to accept this paper.

**Additional Comments On Reviewer Discussion:**

Since all reviewers recognized and positively evaluated the merits of this paper, there was no need for further discussion.

---

### Decision · Program_Chairs · 2025-01-22

Accept (Poster)